# PQBP5/NOL10 maintains and anchors the nucleolus under physiological and osmotic stress conditions

Xiaocen Jin[1,4], Hikari Tanaka [1,4], Meihua Jin[1], Kyota Fujita[1], Hidenori Homma [1], Maiko Inotsume[1], Huang Yong[1], Kenichi Umeda[2], Noriyuki Kodera [2], Toshio Ando [2] & Hitoshi Okazawa [1,3] ✉

Polyglutamine binding protein 5 (PQBP5), also called nucleolar protein 10 (NOL10), binds to polyglutamine tract sequences and is expressed in the nucleolus. Using dynamic imaging of high-speed atomic force microscopy, we show that PQBP5/NOL10 is an intrinsically disordered protein. Super-resolution microscopy and correlative light and electron microscopy method show that PQBP5/NOL10 makes up the skeletal structure of the nucleolus, constituting the granule meshwork in the granular component area, which is distinct from other nucleolar substructures, such as the fibrillar center and dense fibrillar component. In contrast to other nucleolar proteins, which disperse to the nucleoplasm under osmotic stress conditions, PQBP5/NOL10 remains in the nucleolus and functions as an anchor for reassembly of other nucleolar proteins. Droplet and thermal shift assays show that the biophysical features of PQBP5/NOL10 remain stable under stress conditions, explaining the spatial role of this protein. PQBP5/NOL10 can be functionally depleted by sequestration with polyglutamine disease proteins in vitro and in vivo, leading to the pathological deformity or disappearance of the nucleolus. Taken together, these findings indicate that PQBP5/NOL10 is an essential protein needed to maintain the structure of the nucleolus.

The nucleolus is a critical structure for the transcription of ribosomal DNA (rDNA) to ribosomal RNA (rRNA), specifically 45S pre-rRNA, by RNA polymerase I. The nucleolus forms around the tandem repeats of rDNA genes called nucleolar organizing regions (NORs). The 45S pre-rRNA is processed to 18S, 5.8S, and 28S RNA molecules and assembled with 5S rRNA transcribed by RNA polymerase III in the nucleoplasm. Subsequently, these RNA molecules are combined with ribosomal proteins (RPs) in the nucleus and cytoplasm to form 60S and 40S ribosomes. Because the protein translation activity of ribosomes is essential for the maintenance, differentiation, and stress responses of

cells, the nucleolus is a master subcellular structure that regulates cell functions and phenotypes.

Because many components of the nucleolus, such as fibrillarin and nucleolin, have been recognized as intrinsically disordered proteins (IDPs) containing low complexity sequences, the nucleolus is now considered a liquid-liquid phase separation (LLPS) droplet[1–12]. Substructures of the nucleolus include a fibrillar center (FC), a dense fibrillar component (DFC), and a granular component (GC). The FC and DFC are functionally linked to rRNA transcription by RNA polymerase I located at the interface between the FC and the DFC, whereas the functional role of

[1]Department of Neuropathology, Medical Research Institute, Tokyo Medical and Dental University, 1-5-45, Yushima, Bunkyo-ku, Tokyo 113-8510, Japan. [2]Nano Life Science Institute, Kanazawa University, Kakuma-machi, Kanazawa, Ishikawa 920-1192, Japan. [3]Center for Brain Integration Research, Tokyo Medical and Dental University, 1-5-45, Yushima, Bunkyo-ku, Tokyo 113-8510, Japan. [4]These authors contributed equally: Xiaocen Jin, Hikari Tanaka. ✉e-mail: okazawa-tky@umin.ac.jp

the GC remains unclear. The GC region contains the negatively charged molecules nucleophosmin (NPM1) and nucleolin[10,13]. NPM1 homotypic assembly forms the boundary between the nucleolus and nucleoplasm and may contribute to ribosome maturation[5].

Various cell stresses, including DNA damage, temperature change, hypoxia, viral infection, serum starvation, and transcriptional suppression, have been found to affect the morphology and functions of nucleoli[14]. Osmotic stress has been associated with irregular deformation of the nucleolus, although the detailed molecular mechanisms accounting for these morphological changes remain unclear[15]. Osmotic stress has also been reported to affect another nuclear substructure, Cajal bodies, although that study did not investigate nucleolar changes in detail[16]. Osmotic stress has also been shown to trigger phase separation of multivalent IDP proteins[17,18]. Analysis of protein dynamics revealed that misfolded nuclear proteins in the nucleoplasm reversibly entered the nucleolus, especially the GC, under heat shock conditions[10].

For the last 20 years, we have sought to identify key molecules involved in neurodegenerative diseases. For example, HMGB1 was identified by proteome analysis of polyglutamine (polyQ) disease[19], Ku70 by interactome analysis of Huntington's disease (HD)[20], and MARCKS, SRRM2, HMGB1, and YAP by phosphoproteome analysis of Alzheimer's disease pathology[21–24]. In polyQ diseases, we searched for proteins that bind to polyglutamine tract sequences by hypothesizing that such interacting proteins could act as disease modifiers[25]. These analyses resulted in the identification of polyglutamine binding protein 5 (PQBP5), along with polyglutamine binding protein 1 (PQBP1), which was found to be an RNA transcription/splicing-related molecule that regulates neural stem cell proliferation and neuronal synapses[23–28], as well as acting as an intracellular receptor for HIV1 and Tau, regulating inflammatory responses of the cGAS-STING pathway in innate immune cells[29–31]. These analyses also resulted in the identification of polyglutamine binding protein 3 (PQBP3), which has not yet investigated. Interestingly, most of these key molecules (e.g., MARCKS, SRRM2, YAP, and PQBP1) were IDPs.

Nucleolar proteome analysis later identified PQBP5 as a nucleolar protein, which was designated nucleolar protein 10 (NOL10)[32] containing a WD repeat[33]. A meta-analysis of genome-wide association studies revealed that a single nucleotide polymorphism (SNP) in *PQBP5/NOL10* (ID: rs9287719) was a risk factor for prostate cancer[34]. To date, however, gene mutations in *PQBP5/NOL10* have not been extensively analyzed in neurodegenerative or neurological diseases. In addition, GWAS Catalog database has reported that *PQBP5/NOL10* was associated with cardiac troponin levels, arterial stiffness, waist circumference, and liver fibrosis, but not with polyQ diseases (https://www.ebi.ac.uk/gwas/genes/NOL10), although it is unclear whether 46,231 human SNPs linked to *PQBP5/NOL10* gene (https://www.ncbi.nlm.nih.gov/snp/?term=NOL10) have been examined in previous GWAS studies of polyQ diseases[35–38]. Expression profile databases such as Expression Atlas (https://www.ebi.ac.uk/gxa/home) have shown that PQBP5/NOL10 is widely distributed in human/mouse neural and non-neural cells, tissues, and organs. Collectively, current lack of association at the SNP and GWAS levels does not exclude the possibility that PQBP5 is involved in polyQ disease pathology at levels of RNA, protein, protein interaction, protein modification, and/or protein degradation. PQBP5 may also be an IDP whose nucleolar function is associated with nucleolar stresses and, ultimately, with neurodegeneration including polyQ diseases.

Intriguingly, PQBP1, 3, and 5 form a membrane-less substructure in the nucleus, with the IDP-based nuclear or cytoplasmic speckle of PQBP1 being affected morphologically and functionally by mutant polyQ proteins[27] or by cellular stress[39]. In addition, SRRM2, a key IDP in the early stage of neurodegeneration identified, as shown by comprehensive proteome analysis, forms nuclear bodies together with PQBP1[23], suggesting the significance of IDP-based and membrane-less

nuclear substructures, including the nucleolus, in neurodegeneration. Therefore, understanding the role of PQBP5/NOL10 in the morphology and function of the nucleolus in physiological and pathological conditions would lead to investigations of the role of the nucleolus in the pathophysiology of neurodegenerative diseases.

As the first step of this line of research, we investigated the molecular characteristics, nucleolar sublocalization, relationship with other nucleolar proteins, and stress responses of PQBP5/NOL10. Using super-resolution microscopy (SRM) and atomic force microscopy (AFM), this study found that PQBP5/NOL10 is a new type of IDP showing a unique GC-specific distribution in the nucleolus, distinct from that of other nucleolar proteins, and forming the skeletal structure of the nucleolus. During these investigations of the mechanisms by which various cellular stresses affect multiple IDPs, including PQBP5, which we selected as key molecules in neurodegeneration based on multiple comprehensive analyses, we observed that osmotic stress induced differential responses of IDPs, with PQBP5 being the most stable. Unexpectedly, PQBP5/NOL10 was found to act in the nucleolus as a spatial anchor for the assembly of other nucleolar proteins under normal conditions and during recovery from osmotic stress. Deficiency of PQBP5 by knockdown resulted in the absence of nucleoli from cells. Consistently with these analyses of the physiological function of PQBP5, pathological sequestration of PQBP5 to polyQ aggregates in cells and mouse models, which mimic knockdown experiments, resulted in the depletion of nucleolar PQBP5 and the loss or abnormal structures of the nucleolus, suggesting a new aspect of polyQ disease pathology.

## Results
### PQBP5 shows a unique distribution pattern in the nucleolus
Two programs for identifying IDPs, IUPred (https://iupred2a.elte.hu/) and RONN[40], predicted that PQBP5 was an IDP containing an intrinsically disordered region (IDR) at its C-terminal end (Fig. 1a). These programs also predicted that other nucleolar proteins, such as fibrillarin and nucleolin, are IDPs (Fig. 1a). This prediction was confirmed by high-speed AFM, which showed that PQBP5 was composed of a globular region and an unstructured dynamically mobile tail (Fig. 1b, Supplementary Movie 1). AFM showed that part of these PQBP5 molecules were in contact with an RNA molecule, either via a relatively large globular structure transiently formed at the proximal end of the IDR (Fig. 1b, Supplementary Movie 2) or via a small globular structure transiently formed at the terminal end of the IDR (Supplementary Movie 3). These molecular dynamics were illustrated schematically (Fig. 1b, schemes). The transient formation of small structures in the IDP region occurred stochastically and independently of the presence of RNA. These small structures could not contact RNA unless RNA was present nearby (Supplementary Movie 1). To address whether the transient structure is related to interaction, we determined the numbers of times PQBP5 did or did not contact RNA in the presence or absence of the transient structure by evaluating images taken at 1 s intervals (Supplementary Movies 2 and 3). Fischer's exact test suggested that the structure is related to binding (Fig. 1b, tables). These intramolecular structures observed by AFM were well matched with the structure of PQBP5 predicted by AlfaFold (https://alphafold.ebi.ac.uk/) (Fig. 1c).

Morphological analysis of immunostained HeLa cells by SRM (Fig. 1d) revealed that PQBP5 formed a granular dot pattern in the center of the GC, as well as partially overlapping with fibrillarin, a marker of DFC[11] (Fig. 1e, f, g). Fibrillarin was distributed at or around the FC/DFC, as reported[11] (Fig. 1e). Nucleolin was present in the GC, as previously described[10,13], but was predominantly present at the periphery of the nucleolus (Fig. 1e). By contrast, PQBP5 did not show such a peripheral predominance, but was distributed throughout the nucleolus (Fig. 1e). The signal intensity patterns of PQBP5, nucleolin, and fibrillarin differed markedly (Fig. 1f), indicating that their

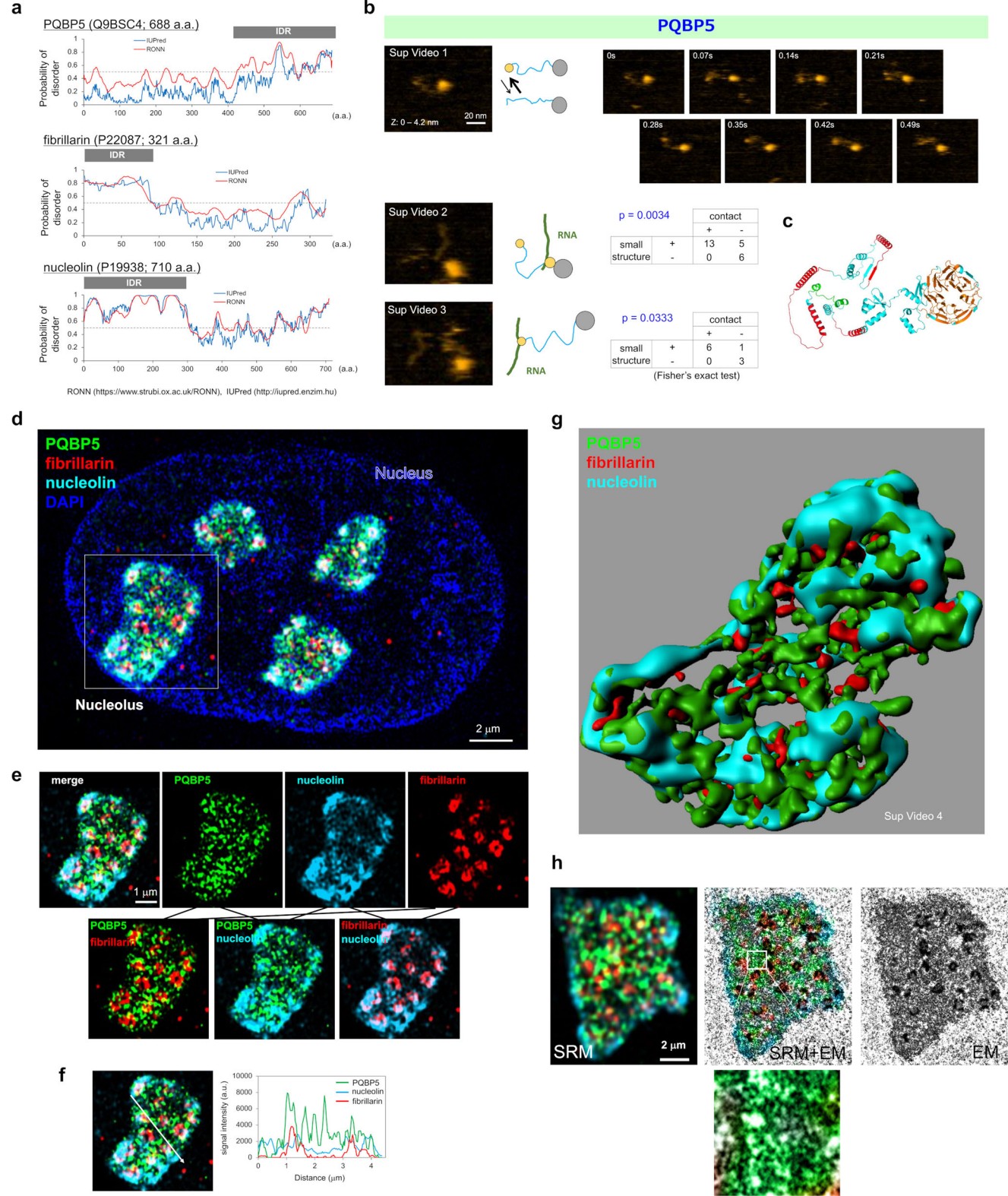

distributions are regulated by more complex biophysical factors. Similar specific distributions of these three proteins have been generally observed in multiple cell types (Supplementary Fig. 1).

The 3D reconstruction of slice images obtained by SRM revealed that PQBP5 formed a type of meshwork that could constitute the skeleton or frame of the nucleolus (Fig. 1g, Supplementary Movie 4). By contrast, nucleolin formed peripheral condensates, and fibrillarin formed granular structures inside the nucleolus (Fig. 1g). Similar

findings were repeatedly confirmed using two SRMs from different companies (Supplementary Figure 2, Supplementary Movies 5–7).

The electron microscopy structures corresponding to the PQBP5 granules in the GC were identified by correlative light and electron microscopy (CLEM) (Supplementary Fig. 3). The unique distribution of PQBP5 signals, as determined by immunocytochemistry and analyzed by SRM, exactly matched the electron-dense meshwork structure composed of the small granules filling the gap space of the GC among

**Fig. 1 | PQBP5 forms the skeletal structure of the nucleolus. a** Prediction of intrinsically disordered regions (IDRs) in human PQBP5, fibrillarin, and nucleolin proteins by two independent programs. The IDR in PQBP5 was predicted at the C-terminus, whereas the IDRs in fibrillarin and nucleolin were predicted at their N-termini. **b** High-speed AFM reveal PQBP5 protein with a globular region and a dynamically moving IDR tail of PQBP5 (upper panel, see Supplementary Movie 1), PQBP5 with a small spherical domain near the globular region interacting with RNA (middle panel, see Supplementary Movie 2) or PQBP5 with a small spherical structure transiently formed at the IDR tail and interacting with RNA (lower panel, see Supplementary Movie 3). The relationship between PQBP5-RNA contact and transient globular structure formation was assessed by two-sided Fisher's exact test. The similar observation was repeated more than three times. **c** Structure of PQBP5 predicted by AlfaFold (https://alphafold.ebi.ac.uk/). **d** Representative super-resolution microscopy (SRM) image of HeLa cells showing the spatial relationship among PQBP5, fibrillarin, and nucleolin proteins at four nucleoli in a nucleus. DAPI shows the area of the nucleus. The nucleolus within the white line square is further shown in Fig. 1e, whereas the other three nucleoli are shown in Supplementary Figure 1. The similar observation was repeated more than ten times. **e** Merged image of the indicated nucleolus showing single and two-protein images. Small PQBP5 dots were distributed homogeneously throughout the nucleolus, distinguishing PQBP5 from nucleolin and fibrillarin. PQBP5 dots were in contact with nucleolin or fibrillarin dots, but did not merge with them. The similar observation was repeated more than ten times. **f** Analysis of signal intensities of PQBP5, nucleolin, and fibrillarin, showing their distinct distribution patterns. **g** 3D image of the three proteins generated by Imaris (see Supplementary Movie 4 for original data). **h** Correlative light and electron microscopy (CLEM) of the three nucleolar proteins. The most electron dense area corresponds to DFC stained with antibody to fibrillarin (red), whereas anti-PQBP5 antibody stained the middle electron dense skeletal structure of the nucleolus (green). The similar observation was repeated more than ten times.

the FC/DFCs determined by electron microscopy (Fig. 1h, Supplementary Fig. 3). As expected, CLEM images of four representative nucleoli revealed that the FC/DFCs determined by electron microscopy matched the fibrillarin staining determined by SRM (Supplementary Fig. 3). CLEM also confirmed that fibrillarin localized the DFC and partial overlapped with PQBP5, which was mainly localized in the GC (Supplementary Fig. 3).

Surface plasmon resonance (SPR) spectroscopy showed a weak interaction between PQBP5 and nucleolin but no interaction between PQBP5 and fibrillarin (Supplementary Fig. 4a). Fibrillarin, however, interacted with nucleolin (Supplementary Fig. 4a). Immunoprecipitation reconfirmed the similar patterns of interaction among PQBP5, nucleolin, and fibrillarin (Supplementary Fig. 4b). These results were useful for considering the interactive relationships among PQBP5, nucleolin, and fibrillarin, together with the results of experiments testing the effects of knockdown (KD) of one protein on another protein and on co-droplet formation, as shown below.

Under normal conditions, NPM1 and Pescadillo ribosomal biogenesis factor 1 (PES1) are distributed in the GC, predominantly in the peripheral region[13,14,20]. Because NPM1 has been reported to circumscribe fibrillarin at the DFC as a ring, rather than being homogeneously distributed in the GC[41,42], we therefore further compared the distribution patterns of these three GC proteins, PQBP5, NPM1, and PES1, in the nucleolus (Supplementary Fig. 5). Our results showed that the distribution patterns of PQBP5, NPM1, and PES1 differed completely (Supplementary Fig. 5). PQBP5 was a major component of the GC, with PES1 and NPM1 surrounding the PQBP5-containing core of the nucleolus, generating the lamellar outer shell of the GC (Supplementary Fig. 5), a finding consistent with previous reports[2,10].

## IDR is essential for nucleolar distribution of PQBP5

To determine the domain or motif responsible for the nucleolar distribution of PQBP5, deletion mutants of PQBP5 were generated (Supplementary Figure 6). PQBP5 contains seven WD repeats (Supplementary Figure 6), which form a globular beta-propeller structure or a WD repeat domain[43], in agreement with the results of high-speed AFM (Supplementary Movies 1–3). The deletion of WD repeats 1 through 4 (Δ1–Δ4 mutants) did not affect the nucleolar distribution of PQBP5 (Supplementary Figure 6). PQBP5-IDR contains coiled coil domains and a NUC 153 domain, which is homologous to a short sequence of NUC 153 and conserved in a novel nucleolar protein family[44], (https://www.ebi.ac.uk/interpro/entry/InterPro/IPR012580/#PUB00016366; https://www.ncbi.nlm.nih.gov/Structure/cdd/PF08159; http://www.ebi.ac.uk/interpro/entry/pfam/PF08159) (Supplementary Fig. 6). Partial to complete deletion of IDR irrespecive of deletion of WD repeats (Δ5–Δ8 mutants) changed the distribution pattern of PQBP5 from the nucleolus to the cytoplasm (Supplementary Fig. 6). A comparison of the Δ4 and Δ5 mutants indicated that the NUC 153 domain is especially critical for the nucleolar distribution of PQBP5 (Supplementary Fig. 6).

## PQBP5 is a dominant determinant essential for nucleolus formation

Knockdown experiments of PQBP5, nucleolin, and fibrillarin were performed to evaluate the possible role of PQBP5 as a framework for nucleolus formation (Fig. 2). Transient transfection of a pAV-U6-GFP vector expressing shRNA for one of the three proteins (PQBP5, nucleolin or fibrillarin) (Fig. 2a) was performed to evaluate the effect of depletion of each of these proteins (Fig. 2b) on the distribution of the other two (Fig. 2c). Knockdown of PQBP5 markedly reduced the nucleolar concentrations of nucleolin and fibrillarin and induced their homogeneous distribution in the nucleus (Fig. 2c, upper panels), but did not reduce the total amounts of nucleolin and fibrillarin proteins within a short period of time (Fig. 2b). Knockdown of fibrillarin did not affect the nucleolar concentrations of PQBP5 and nucleolin (Fig. 2c, middle panels), whereas knockdown of nucleolin reduced the nucleolar concentration of fibrillarin but did not affect that of PQBP5 (Fig. 2c, lower panels). Original images of shRNA-mediated knockdown are also shown (Supplementary Fig. 7).

Immunoelectron microscopy testing the effect of knockdown of one of the three proteins on the number and shape of the nucleoli further indicated that PQBP5 has the largest effect on nucleolus formation among the three proteins (Supplementary Fig. 8). Western blotting showed that PQBP5, fibrillarin, and nucleolin were similarly knocked down (Supplementary Fig. 8a). A low level of expression of PQBP5-shRNA-EGFP in cells (gold particles of anti-EGFP antibody against shRNA-EGFP < 80/μm$^2$ visual field) resulted in nucleolar deformation, whereas high levels of PQBP5-shRNA-EGFP (gold particles > 80/μm$^2$ visual field) mostly resulted in the absence of all nucleoli (Supplementary Fig. 8a, third row panels). The number of nucleolus-negative cells was highest in the presence of PQBP5-shRNA-EGFP expression (Supplementary Fig. 8b).

Nucleoli in HeLa cells expressing PQBP5-shRNA have features such as irregular shape, low electron density, and DFC disappearance (Supplementary Fig. 8c). "Abnormal cells" were therefore defined as cells with nucleoli having two or more of these abnormal features. The effect of knockdown of each protein on the percentage of abnormal cells, irrespective of their expression levels, was therefore tested (Supplementary Fig. 8d). The frequencies of abnormal cells, as shown by the shape of the nucleoli, were higher in gold particle-positive cells with shRNA-EGFP expression for PQBP5-KD than with shRNA-EGFP for nucleolin- or fibrillarin-KD (Supplementary Fig. 8d).

Collectively, these findings suggest that PQBP5 is the most dominant regulator among the three IDPs (PQBP5, fibrillarin, and nucleolin) (Fig. 2d), and indicate that PQBP5 is essential for nucleolus formation.

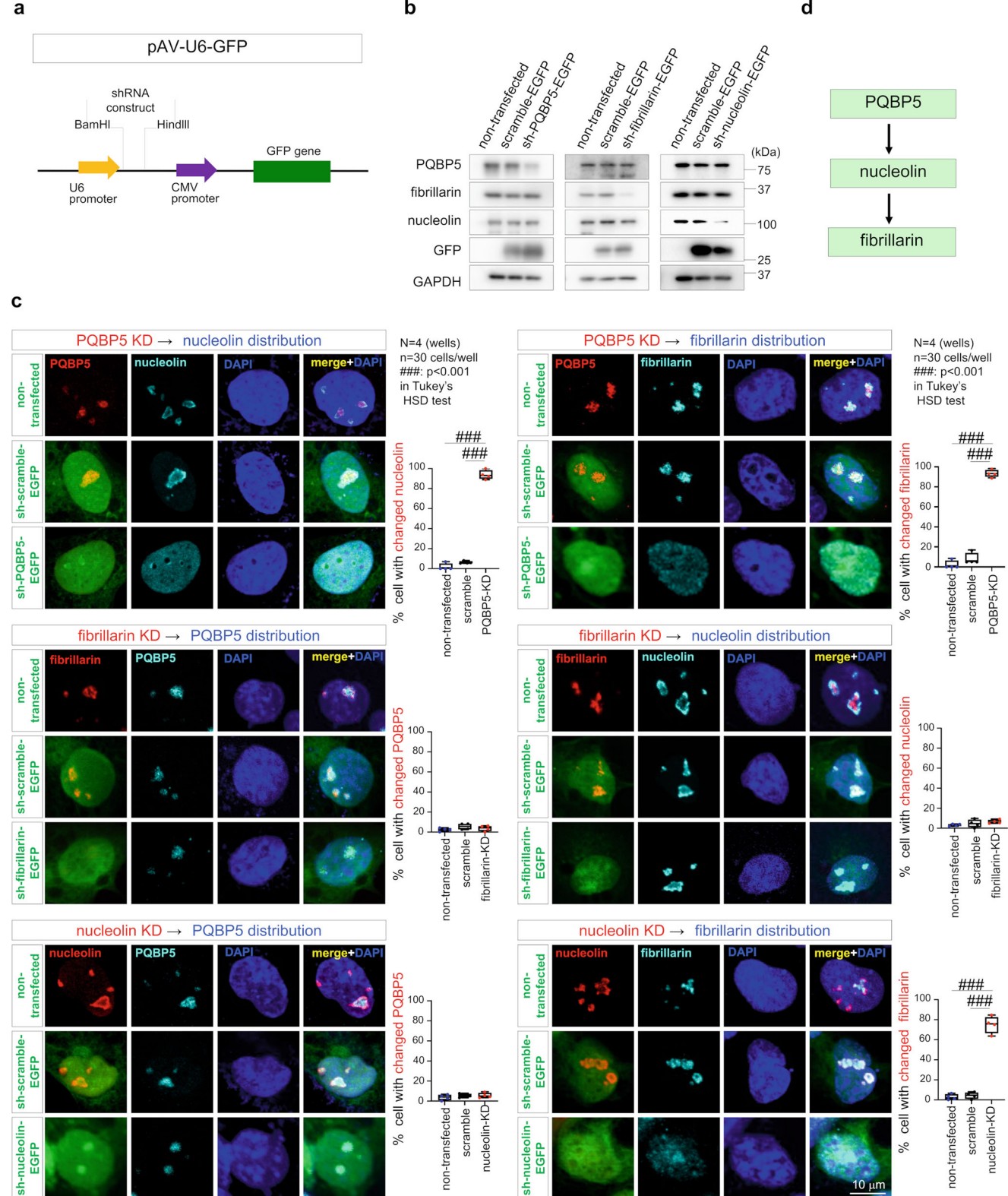

## PQBP5 anchors and maintains the nucleolus under osmotic stress conditions

Our preliminary experiments testing the effects of various cellular stresses on candidate IDPs, selected from our comprehensive screenings as key molecules in neurodegeneration, found that osmotic stress differentially modified distributions of PQBP5 and partner molecules. In addition, because dehydration frequently occurs in elderly people[45,46] and during hyperglycemic episodes in patients with

diabetes mellitus[47–49] and because dehydration-induced delirium[50,51] and diabetes melitus[52] are risk factors for dementia, the function of PQBP5 in the nucleolus was further assessed under osmotic stress conditions. HeLa cells were incubated for 20 min under hypo- or hyper-osmolar conditions, after adjustment with sorbitol, and cultured under normal osmotic conditions (Fig. 3a). Sequential samplings and confocal microscopy showed that nucleolin and fibrillarin, but not PQBP5, dispersed immediately in culture medium at 220 or 500/

**Fig. 2 | PQBP5 is essential for nucleolar assembly of fibrillarin and nucleolin.**
**a** Structure of the PQBP5, fibrillarin and nucleolin knockdown vectors. The target sequences with the loop region were subcloned between the BamHI and HindIII sites. **b** Western blot analysis of HeLa cells transfected transiently with plasmids expressing PQBP5, fibrillarin, and nucleolin shRNAs. These shRNAs reduced the levels of expression of PQBP5, fibrillarin, and nucleolin to nearly 30% of the levels in untransfected cells. The similar experiment was repeated three times.
**c** Immunocytochemistry showing the levels of expression of each target protein in cells transfected transiently with plasmids expressing PQBP5, fibrillarin, and

nucleolin shRNAs. The expression of each target protein was almost completely suppressed in transfected cells. Knockdown of PQBP5 induced dispersion of fibrillarin and nucleolin. Fibrillarin knockdown did not affect the distribution of the other proteins. Nucleolin knockdown affected the distribution of fibrillarin but not of PQBP5. The box plot shows median, 25–75th percentile, and whiskers representing data outside the 25–75th percentile range. Tukey's HSD test was used for multiple comparisons. $P < 0.0001$ in all comparisons by GraphPad Prism.
**d** Hierarchy of PQBP5, nucleolin, and fibrillarin determined by knockdown experiments.

700 mOsm (Fig. 3b). During the recovery phase, nucleolin and fibrillarin promptly reassembled in an area that had retained PQBP5 (Fig. 3b). Quantitative analyses of the signal intensity of nucleolin and fibrillarin also showed that both had dispersed to the nucleoplasm in response to osmotic stress (Fig. 3c).

SRM yielded similar results, showing the dispersion of the nucleolar proteins under osmotic stress conditions and their reassembly in the nucleolus during recovery (Supplementary Fig. 9). Although the signals of dispersed fibrillarin appeared relatively low in single slice images (140 nm) of SRM, the maximum intensity projection method based on the maximum number of voxels in all projections revealed that fibrillarin dispersed under both hypo- and hyper-osmolar conditions (Supplementary Fig. 10).

### Relationship of nucleolar protein assembly, higher order chromatin structure, and Pol I transcription

Nucleoli can also be evaluated by DAPI staining, which can distinguish between high and low-density areas of chromatin fibers. We therefore compared DAPI and nucleolar protein staining patterns in normal HeLa cells (Supplementary Fig. 11). Nucleoli stained by antibodies to fibrillarin and nucleolin tended to be located in DAPI negatives area surrounded by high DAPI signals, although such areas were not always nucleoli and vice versa. These findings indicated that DAPI staining could not replace nucleolar protein staining for evaluation of nucleoli.

DAPI staining also showed that knockdown of PQBP5 abrogated the assembly of nucleolar proteins but did not alter chromatin fiber-free spaces for generation of nucleolus in the short term (Fig. 2). Similar findings were observed when dispersion of nucleolar proteins was assessed after changes in osmotic conditions (Fig. 3). Higher order chromatin structures kept the positions for nucleolar proteins to reassemble to form nucleoli, suggesting that the order of nucleolus formation consisted initially of rDNA exposure followed by the assembly of nucleolar proteins, within PQBP5 playing an essential role in the latter. This hypothesis is consistent with the current view that nucleoli are formed at nucleolus organizing regions (NORs), where higher order chromatin structures such as chromatin fibers are relaxed, exposing rDNAs to the RNA polymerase I (Pol I) transcription machinery[53,54]. Determining the chronological relationship between higher order chromatin structure and nucleolar proteins requires extensive investigation.

Inhibition of Pol I-driven transcription can cause re-localization of nucleolar proteins into the nucleoplasm/cytoplasm and loss of nucleolar integrity. Therefore, the consequences of depleting PQBP5 may be due to the loss of 47S rRNA transcription, which could be independent from any structural role of PQBP5. Because PQBP5 binds RNA, the effects of PQBP5 loss on 47S rRNA transcription should be evaluated; that is, it is important to assess whether PQBP5 affects the nucleolus directly or indirectly via 47S rRNA. We therefore examined 47S rRNA, PQBP5 protein, and GAPDH protein levels (Supplementary Fig. 12) in parallel with PQBP5 knockdown (Fig. 2). Forty-eight hours after shRNA transfection, when the nucleolus was changed morphologically (Fig. 2), PQBP5 protein was decreased in western blot while 47S rRNA, which was evaluated by qPCR, was unchanged (Supplementary Fig. 12). At 72 h, however, both PQBP5 protein and 47S rRNA were decreased (Supplementary Fig. 12). These results collectively

suggested that PQBP5 affected the nucleolus directly rather than indirectly via 47S rRNA.

### LLPS of PQBP5 reconstructs the nucleolus in vitro

PQBP1, another protein that binds to polyglutamine tract sequences, was one of the first IDPs that showed the distribution of LLPS droplets in the nucleoplasm outside the nucleolus[27]. In addition, PQBP1 formed lamellar structure droplets with another nuclear protein, ataxin-1 (Atxn1)[27,55]. These findings suggested the need to evaluate the biophysical features of PQBP5, nucleolin, and fibrillarin in LLPS and to analyze their differences and mutual relationships.

Because AFM revealed an interaction between PQBP5 and rRNA (Fig. 1b) and because PQBP5-KD impairs assembly of fibrillarin and nucleolin (Fig. 2), we hypothesized that the presence of rRNA would influence droplet formation of PQBP5 or multiple IDPs mediated by PQBP5. Therefore, droplet formation by PQBP5, nucleolin and fibrillarin, each at a concentration of 250 nM, was examined in the absence or presence of 2.5 μg/ml HeLa cell rRNA (Fig. 4a). As expected, droplet formation by PQBP5, nucleolin, and fibrillarin was markedly increased by the addition of rRNA (Fig. 4a). In the absence of rRNA, only PQBP5 formed a few droplets, while nucleolin never and fibrillarin rarely formed droplets. The addition of rRNA enhanced PQBP5 droplet formation and increased the sizes of these droplets (Fig. 4a). The addition of rRNA also increased the number and sizes of fibrillarin droplets, although these enlarged droplets were smaller than the PQBP5 droplets (Fig. 4a). Moreover, the fibrillarin droplets were unstable and disappeared 9 h after formation (Fig. 4a). Even in the presence of rRNA, only a very few nucleolin droplets were observed (Fig. 4a).

In the presence of HeLa cell rRNA (Fig. 4a), droplet formation was also assessed in mixtures of two or three proteins (Fig. 4b). PQBP5 or fibrillarin alone did not enhance nucleolin droplet formation at 3 hours (Fig. 4b), whereas nucleolin cooperatively formed triple protein droplets in the presence of both PQBP5 and fibrillarin (Fig. 4c). Interestingly, fibrillarin formed a ring around nucleolin in vitro (Fig. 4c), the reverse of the in vivo spatial relationship, in which fibrillarin forms DFC inside nucleolin in the nucleolus. The physical characteristics of the background fluid and additional co-existing IDPs could affect droplet formation of two IDPs by changing the Helmholtz free energy[56]. Therefore, opposite relationships may occur in droplet assay buffer in vitro and nucleoplasm in vivo, including a number of other IDPs.

The resistance of each of these protein droplets to osmotic changes was also examined (Fig. 4d). PQBP5 droplets were stable, whereas fibrillarin droplets were sensitive to osmotic changes (Fig. 4d), in agreement with the results of confocal microscopy and SRM (Fig. 3, Supplementary Figs. 9 and 10).

### The three nucleolar proteins differ in physicochemical properties

To investigate the physicochemical features of IDPs that could explain the relative stability of PQBP5 in comparison to fibrillarin and nucleolin, we performed thermal shift assays in normal and abnormal osmotic conditions. Thermal shift assays based on dye labeling of hydrophobic amino acids have been used to investigate structural changes in proteins in response to altered temperatures. Assays using recombinant PQBP5, nucleolin, and fibrillarin proteins showed that

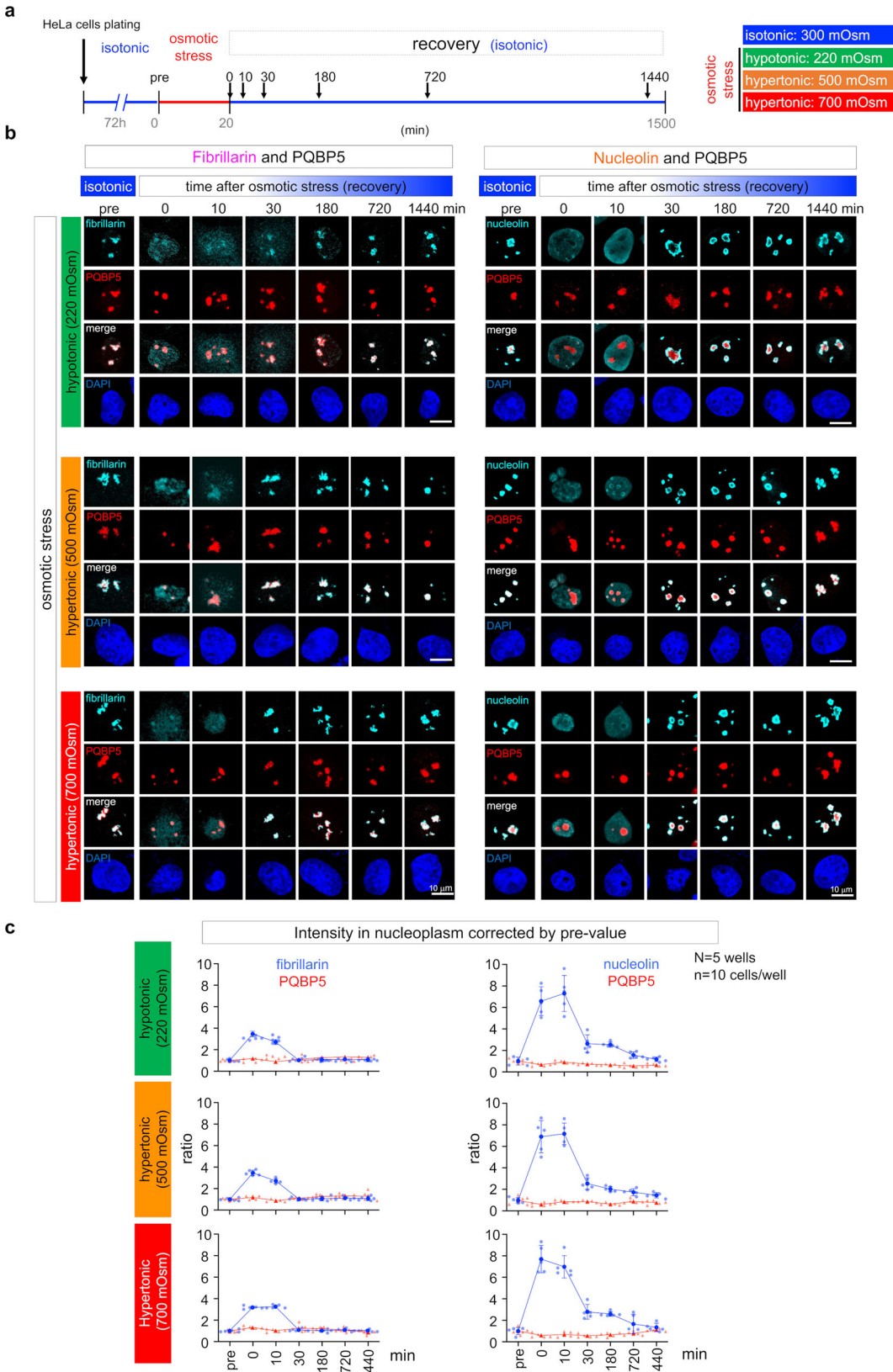

**Fig. 3 | PQBP5 anchors the nucleolus under osmotic stress conditions.**
**a** Experimental protocol for the induction of osmotic stress. **b** Changes in the nucleolar distribution of nucleolar proteins during and after osmotic stress, as shown by the relationships of PQBP5 with fibrillarin and nucleolin in a single cell. HeLa cells subjected to three types of osmotic stress showed dispersion of fibrillarin and nucleolin, but retention of PQBP5, in the nucleolus. During the recovery phase, fibrillarin and nucleolin returned to the area of PQBP5 retention. **c** Quantification of signal intensity/area (signal density) in the nucleus outside the nucleolus during and after osmotic stress, as shown by the relationships of PQBP5 with fibrillarin and nucleolin. Fibrillarin and nucleolin dispersed from the nucleolus to the nucleoplasm during stress, whereas PQBP5 did not. Data are presented as mean values ± SEM.

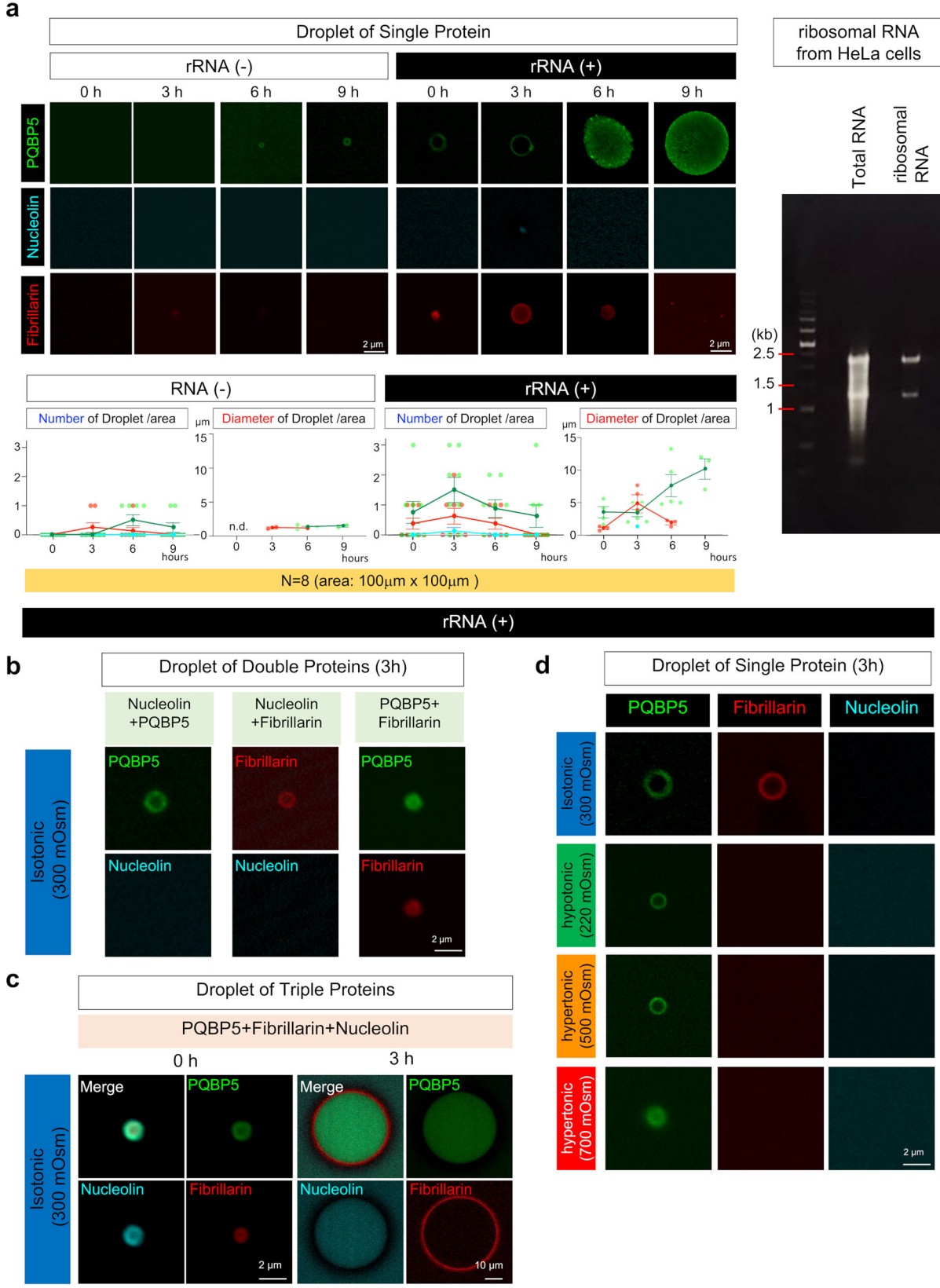

nucleolin was immediately labeled with thermal shift dye at 25 °C, whereas fibrillarin was less labeled and PQBP5 was hardly labeled (Fig. 5a, left panels). The protein structures predicted by AlfaFold (https://alphafold.ebi.ac.uk/) indicated that the total length of degenerated regions was longest in nucleolin, moderate in fibrillarin, and shortest in PQBP5 (Fig. 5b). The other regions consisted almost completely of α-helices (Fig. 5b).

Previous studies of the effects of changing temperature from 150 °C to 550 °C on multiple α-helical structures indicated that these structures were stable at 100 °C[57]. Therefore, the signal changes

**Fig. 4 | Biophysical characteristics underlying the anchor function of PQBP5.**
**a** Purified PQBP5, nucleolin, and fibrillarin proteins were fluorescently labeled at different wavelengths, and the frequency and size of droplets were investigated in the absence or presence of ribosomal RNA. The lower graphs show quantitative analyses of eight chips (observed area 100 μm × 100 μm). The right agar gel confirms the quality of ribosomal RNA after purification. In the absence of RNA (left panels and graphs), small numbers of PQBP5 droplets were observed 6 and 9 h after mixing the fluorescent protein with buffer. A few fibrillarin droplets were observed at 3 and 6 h, whereas no nucleolin droplets were detected. In the presence of RNA (right panels and graphs), PQBP5 droplets appeared immediately after mixing, with the number peaking at 3 hours, and the sizes continuing to increase for up to 9 h. The number and size of fibrillarin droplets both peaked at 3 h, but these droplets disappeared thereafter. A few nucleolin droplets were observed only at 3 h. Data are

presented as mean values ± SEM. **b** Mixtures of two nucleolar proteins under isotonic conditions. Neither PQBP5 nor fibrillarin cooperatively formed droplets with nucleolin (left and middle panels), whereas PQBP5 and fibrillarin formed droplets cooperatively (right panels). The similar experiment was repeated more than five times. **c** Formation of co-droplets by the three nucleolar proteins. Initially, PQBP5 and nucleolin surrounded fibrillarin, but their lamellar relationship was reversed at 3 h. The similar experiment was repeated more than five times. **d** Effect of osmolarity on droplet formation by the three nucleolar proteins. PQBP5 droplets were most resistant to changes in osmolarity. Fibrillarin droplets did not form under hypertonic and hypotonic conditions, whereas few nucleolin droplets were observed under all conditions. The similar experiment was repeated more than five times.

observed from 25 °C to 100 °C in this experiment were considered to reflect structure changes of IDP regions (IDR). The signals of PQBP5, nucleolin, and fibrillarin that have undergone complete degeneration, except for the α-helices, may be high at 25 °C, depending on the total length of the degenerated regions. These signals therefore would not be increased by increasing the temperature to 100 °C.

These assays were subsequently modified by altering the osmolarity of the reaction solution. The signals were increased under both hypotonic and hypertonic conditions at all temperatures, with nucleolin showing a greater increase than fibrillarin (Fig. 5a, right panels). The variation in the fluorescence of PQBP5 among isotonic, hypotonic, and hypertonic conditions (8779.6 fluorescence units) was similar to or slightly higher than the background change of the buffer (5218.0 fluorescence units), while the variation in osmotic stress was far higher in the fluorescence of nucleolin (346,818.9 fluorescence units) and fibrillarin (157,697.4 fluorescence units) (Fig. 5a, right panels), indicating that significant structural changes were not induced in PQBP5-IDR by hypo- or hyperosmolarity and that PQBP5 is structurally most stable among these three IDPs under osmotic stress conditions. In addition, these results indicated that PQBP5 is structurally most stable among the three IDPs under thermal stress conditions in vitro (Fig. 5a, left panels).

## PQBP5 deprivation by polyglutamine disease protein impairs the nucleolus

PQBP5 was originally identified as a protein that bound to the polyQ-tract sequence associated with polyQ disease[25], with the results of this study showing that PQBP5 plays an essential role in nucleolar maintenance under physiological and stress conditions (Figs. 1–4). PQBP5 is essential for maintaining the structure of the nucleolus, although PQBP5 expression in the nucleolus could be reduced following sequestration of interacting proteins in inclusion bodies resulting from the generally accepted pathomechanism of polyQ diseases predicted by the "sequestration hypothesis". The sequestered proteins become less mobile, as their partner disease proteins are insoluble[58–62]. These findings prompted us to hypothesize that interactions between PQBP5 and polyQ disease proteins might impair the morphology of the nucleolus, which can lead ultimately to nucleolar dysfunction. To test this hypothesis, we first investigated the interactions between PQBP5 and polyQ disease proteins and addressed the resultant effects.

Normal and mutant Atxn1, the protein causing spinocerebellar ataxia type 1 (SCA1)[63], and Huntingtin (Htt), the protein causing Huntington's disease (HD)[64], were transiently expressed in U2OS cells (Fig. 6). Mutant Atxn1 and mutant Htt-Exon1 proteins, but not normal Atxn1 or normal Htt-Exon1 protein, sequestered PQBP5 and dispersed nucleolin and fibrillarin (Fig. 6, left panels). Quantitative analysis confirmed that PQBP5 signals were sequestered to areas containing mutant Atxn1 or mutant Htt-Exon1 aggregates and that the nucleolar signals of nucleolin and fibrillarin were reduced (Fig. 6, right graphs). Such tendencies were generally observed in visual fields at low magnification (Supplementary Fig. 13).

Similar reductions in PQBP5, nucleolin, and fibrillarin were observed in the nucleoli of striatal neurons of mutant Htt-KI mice (Hdh^{Q111} knockin mice) and of cortical neurons or Purkinje cells of mutant Atxn1-KI mice (Sca1^{154Q/2Q} knock-in mice) (Fig. 7a). Electron microscopy showed that the features of the nucleoli in some of the neurons of the two KI mice were abnormal (Fig. 7b), similar to findings in cultured PQBP5-KD cells (Supplementary Fig. 8c, d).

SPR (Supplementary Fig. 14) analyses of interactions between polyQ disease proteins and full-length PQBP5 (FL-PQBP5) confirmed that FL-PQBP5 interacted with both normal and mutant forms of Htt-Exon1 and Atxn1 proteins in vitro. Co-immunoprecipitation analyses using U2OS cells transiently expressing FLAG-FL-PQBP5 and Myc-Htt or Myc-Atxn1 basically supported these interactions (Supplementary Fig. 15a, b), except for the in vivo interaction between FL-PQBP5 and normal Htt. The latter result was due probably to the localization of FL-PQBP5 in the nucleoli while normal Htt was distributed in the cytoplasm (Fig. 6). Normal Atxn1, however, co-localized with FL-PQBP5 in the nucleoplasm (Fig. 6).

Co-immunoprecipitation analyses of polyQ disease proteins with PQBP5 deletion mutants lacking different numbers of WD repeats[65,66] or the NUC 153 domain[44] (Supplementary Fig. 15) suggested that the WD repeats of PQBP5 were necessary for these interactions. Interestingly, although mutant Atxn1 had a slightly higher binding affinity (Kd) to PQBP5 than normal Atxn1 (Supplementary Fig. 14), the amounts of PQBP5 co-immunoprecipitated differed only slightly between normal and mutant Atxn1 (Supplementary Fig. 15). On the other hand, the Kd of normal Htt-exon1 to PQBP5 was an order of magnitude higher than that of mutant Htt-exon1 (Supplementary Fig. 14), with the amounts of PQBP5 co-immunoprecipitated differing markedly (Supplementary Fig. 15). These findings suggested that sequestration of PQBP5 in inclusion bodies composed of insoluble or less mobile mutant polyQ disease proteins could be the cause of PQBP5 reduction in the nucleolus. In addition, subsequent processes, such as degradation of the complexes with mutant proteins, would contribute to the reduction of PQBP5.

## Discussion

The present study was designed to reveal the physiological and pathological functions of PQBP5, a protein originally identified as a protein binding to polyQ-tract sequences by the yeast two-hybrid method[25]. We originally hypothesized in 1998 that proteins binding to polyQ sequences could be modifiers of polyQ diseases[25]. Before evaluating the pathological function of PQBP5, however, it was necessary to characterize its biophysical properties and functions. To characterize the features of IDPs, it is first necessary to determine their biophysical properties in response to altered conditions, such as heat and osmolarity. The present study showed that PQBP5 is relatively stable in comparison with other IDPs in the nucleolus, indicating the role of PQBP5 in anchoring the nucleolus. PQBP5 knockdown reduced the nucleolar concentrations of nucleolin and fibrillarin before the decrease of 47S rRNA, indicating that PQBP5 affected the nucleolus

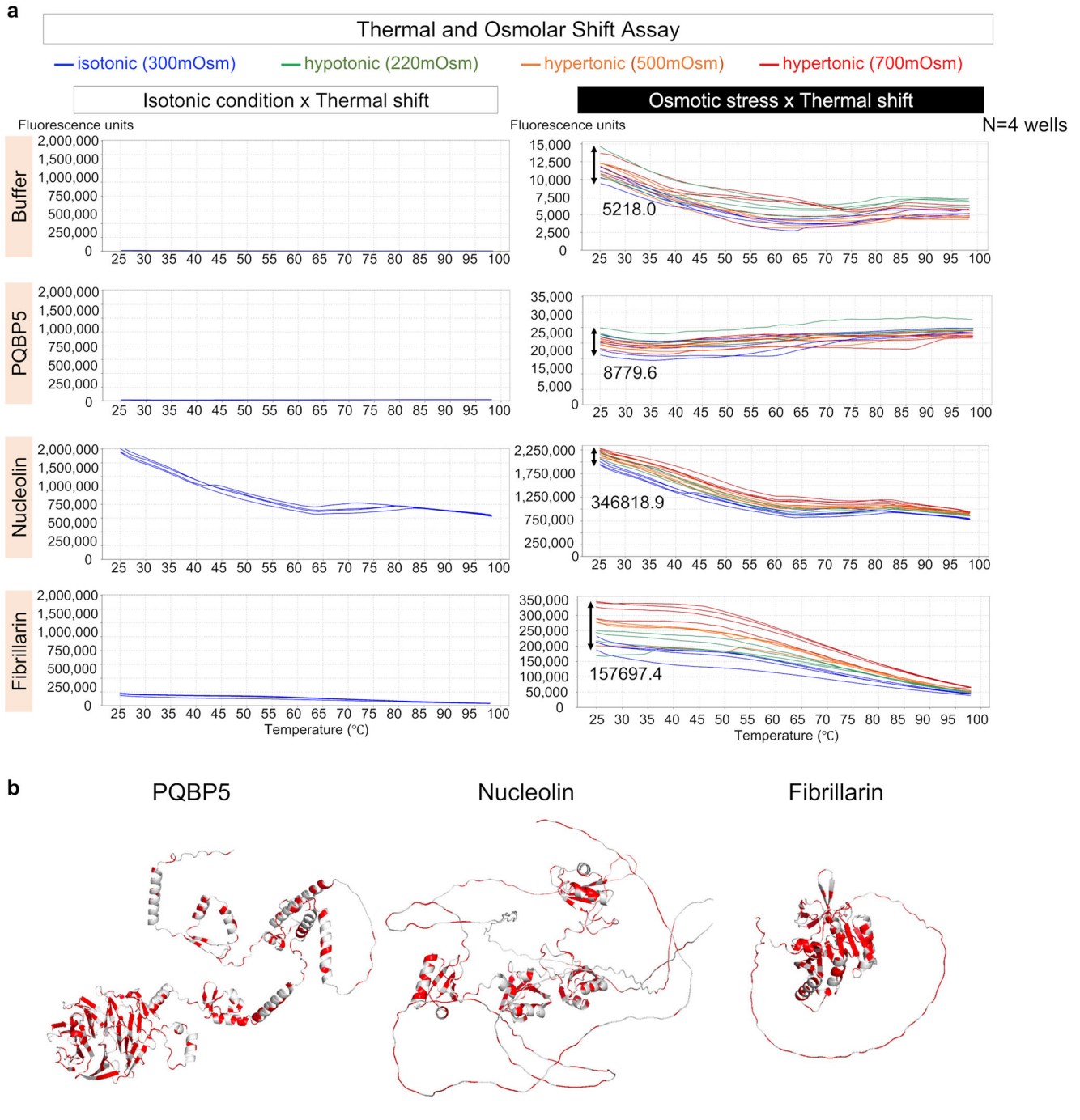

**Fig. 5 | Thermal and osmotic shift assays reveal differences in the physico-chemical properties of the three nucleolar proteins. a** Thermal shift assay (left panels). Florescence signal intensities of the three nucleolar proteins at an osmolarity of 300 mOsm while changing the temperature from 25 °C to 99 °C. The frequency of exposure of hydrophobic amino acids at the outer surfaces of PQBP5, nucleolin, and fibrillarin differed markedly among the three proteins. Thermal shift assays under multiple osmotic conditions (right panels) showing that the three nucleolar proteins differed in their susceptibilities to changes in osmolarity. **b** Structures of PQBP5, nucleolin, and fibrillarin predicted by AlfaFold2.

directly rather than indirectly via 47S rRNA. We therefore investigated the role of PQBP5 in the pathology of polyQ diseases. Consistent with the sequestration hypothesis, we found that PQBP5 was absent from polyQ protein inclusion bodies in polyQ diseases. These findings suggested that conditions observed in polyQ diseases are similar to those observed following PQBP5-KD, with depletion of PQBP5 leading to the loss or marked deformity of the nucleolus in neurons.

The nucleolus is an essential subcellular and suborganelle structure in eukaryotic cells and animal bodies[67–70]. The nucleolus is now recognized as an LLPS-based structure[1–12], with several studies investigating the structure, component molecules, and functions of the nucleolus[71]. For example, fibrillarin was shown to be involved in sorting rRNAs from the FC/DFC border, where rRNAs are synthesized[71,72], to the DFC[11]. NPM1, a component protein of the GC, is considered critical for nucleolar structure[5,12,73,74]. In addition, many studies have investigated the effects of various types of cell stresses on the morphology and function of the nucleolus[14]. For example, a comprehensive and quantitative proteome analysis revealed the

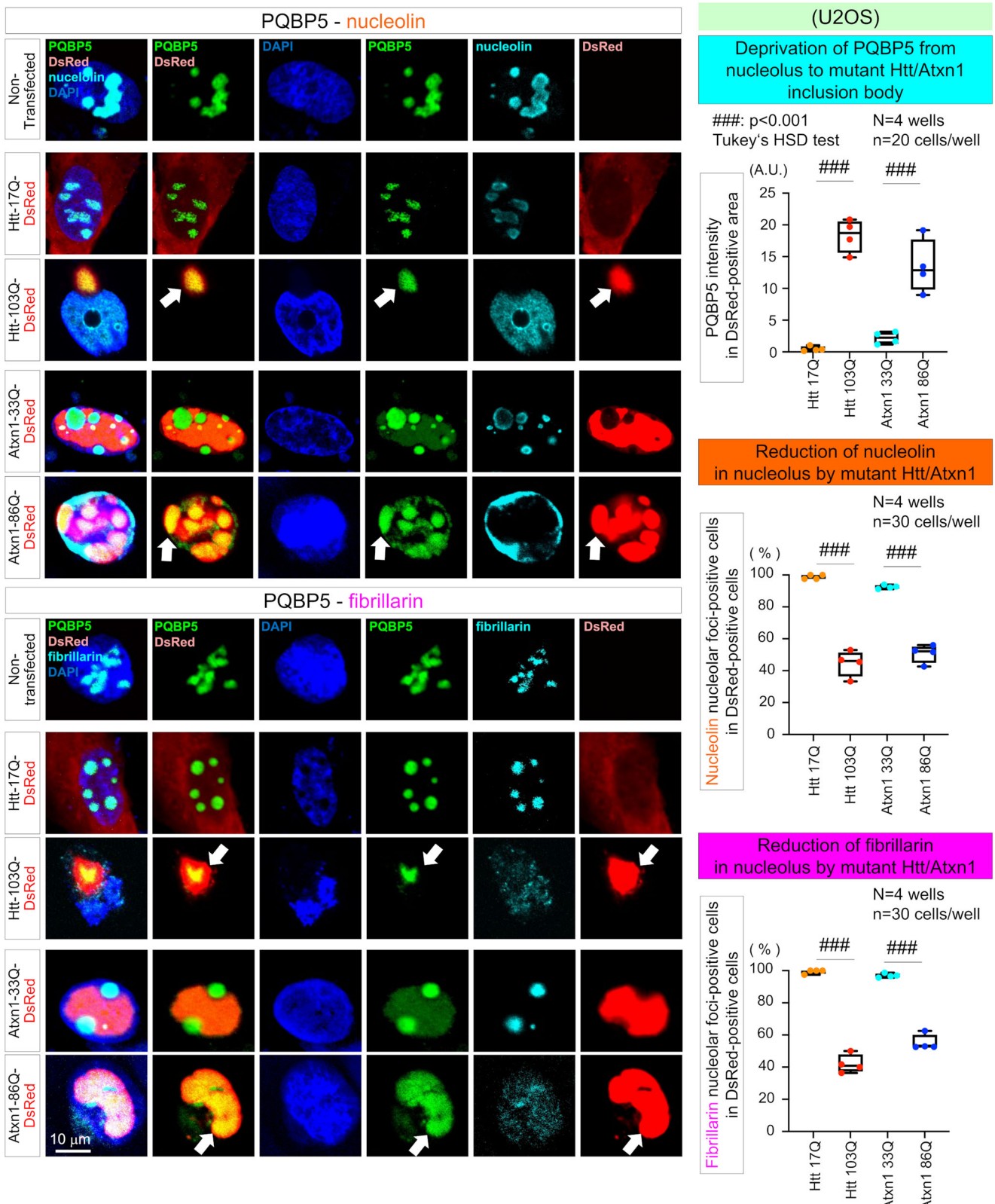

localization shift of proteins between nuclear and nucleolar fractions induced by DNA damage[75,76], while attention was not focused on non-shifted proteins. In response to heat shock, Chromobox 2 (CBX2), a protein component of the polycomb group complex, is concentrated in the nucleolus; however, CBX2 is not regarded as an anchor of the nucleolus, with the phenotype considered as a protective response for CBX2[77]. To date, however, the functional roles of all nucleolar components, including PQBP5 and PQBP3, have not been investigated completely. In addition, molecule-based responses under osmotic stress have not been determined sufficiently. Therefore, some molecular mechanisms required for the maintenance of nucleoli under physiological conditions and for the assembly of multiple proteins and RNAs have not been completely determined.

**Fig. 6 | Sequestration of PQBP5 in inclusion bodies results in the dispersion of nucleolar proteins.** Mutant Htt and Atxn1 were expressed in U2OS cells, and their effects on PQBP5 and other nucleolar proteins were investigated. Upper panels show sequestration of PQBP5 to cytoplasmic inclusion bodies of mutant Htt and nuclear inclusion bodies of mutant Atxn1. Expression of mutant Htt and mutant Atxn1 resulted in the dispersion of nucleolin to the nucleoplasm. Lower panels show the effect of PQBP5 sequestration on nucleolar assembly of fibrillarin. Right graphs show quantitative analyses of the signal intensity of PQBP5 sequestered to polyglutamine disease protein inclusions (upper), the percentage of nucleolin nucleolar foci-positive cells among transfected cells (middle), and the percentage of fibrillarin nucleolar foci-positive cells among transfected cells (lower). The box plot shows median, 25–75th percentile, and whiskers representing data outside the 25–75th percentile range. Tukey's HSD test was used for multiple comparisons. *P* = 0.0002 in Atxn1-33Q vs Atxn1-86Q comparison of PQBP5 intensity in DsRed-positive area and *P* < 0.0001 in other comparisons by GraphPad Prism.

By contrast, nucleolar dysfunction has been implicated as a pathological domain of polyQ diseases. For example, expanded CAG repeat-containing RNAs have been found to directly interact with nucleolin, preventing binding to the rRNA promoter and impairing nucleolar function[78]. Conditional knockout of the RNA polymerase I-specific transcription initiation factor IA (TIF-IA) has been reported to cause Huntington's disease-like striatal degeneration[79]. Moreover, dysfunction of upstream binding factor-1 (UBF-1) has been reported to contribute to HD pathology in mouse and cellular models by reducing ribosomal DNA (rDNA) transcription[80].

The present study showed that PQBP5, which was originally identified as a protein binding to the polyQ-tract sequence[25,26], is the key molecule that maintains and regulates nucleolar structure under both normal and pathological conditions. Without PQBP5, other nucleolar proteins such as fibrillarin and nucelolin, could not assemble to form the nucleolus, and PQBP5 was found to function as an anchor for the reassembly of fibrillarin and nucleolin after osmotic stress. This study also expanded knowledge about the super-resolution structure and biophysics of PQBP5, showing that it is a self-assembling IDP that forms a meshwork structure in the GC. Presumably, this protein could constitute the skeleton of the nucleolus due to its relatively high biophysical stability under thermal and osmotic changes, and could function as an anchor for the reassembly of some nucleolar proteins after nucleolar stress.

The present study also showed that the indispensable physiological function of PQBP5 could be hindered by its pathological sequestration to inclusion bodies of mutant polyQ proteins and by resultant depletion of nucleolar PQBP5. Although PQBP5 was found to interact with normal polyQ protein, nucleolar PQBP5 was found to be selectively depleted in the presence of mutant polyQ proteins presumably due to immobilization to fibrillary inclusion bodies[60–62]. Many previous studies indicated interaction-based sequestration of physiological nuclear proteins into nuclear or cytoplasmic inclusion bodies of mutant polyQ disease proteins; this sequestration, however, does not occur with normal polyQ disease proteins due to the lack of sticky immobilization, even if interactions occur[81–83]. Similarly to the general concept, we found that the forced expression of mutant polyQ disease proteins in cultured cells and the physiological-level expression of mutant polyQ disease proteins in KI mice leads to nucleoli with abnormal morphology, findings not observed with normal polyQ disease proteins. The sequestration of PQBP5 by mutant polyQ disease proteins suggested that the instability of the nucleolus due to a deficiency in an anchor protein could lead to neurodegeneration. Further investigations are needed to confirm this hypothesis and to reveal the detailed mechanism underlying the association between PQBP5 and mutant polyQ diseases.

Unexpectedly, high-speed AFM showed that PQBP5 interacted with contaminating RNA in protein samples derived from *Escherichia coli*. Therefore, we evaluated the species and amount of the interacting RNA. Agar gel electrophoresis of co-purified RNAs showed faint bands corresponding to 23S and 16S ribosomal RNA, as well as faint low molecular smears around 1.0 kb and below 0.5 kb (Supplementary Fig. 16a). RT-PCR did not detect mRNA or tRNA abundantly expressed in *E. coli* (Supplementary Fig. 16b), indicating that the RNA molecules would be small bacterial rRNAs such as 5S rRNA, the counterpart of 5.8S mammalian rRNA, degraded rRNA or spliced peptide fragments of tRNA (tiRNA)[84,85]. Based on their relative abundance, we assumed that the RNA presumably originated from rRNA. Consistent with this hypothesis, we found that the addition of total rRNA enhanced droplet formation of PQBP5 strongly and that of fibrillarin weakly.

In these high-speed AFM experiments, the protein samples contained 2.5 ng/µl RNA (or 9.7 nmol/l based on mean size) and the PQBP5 concentration was 500 nmol/L, a ratio of about 1:50. Under these conditions, we observed fourteen PQBP5 molecules, including two bound and one un-bound to RNA. The dynamics of PQBP5-RNA interactions should be further investigated, as should the identification of the RNA species interacting with PQBP5. Determination of the mechanisms by which various nucleolar proteins interact with different types of RNAs or proteins, including those that are associated with neurodegeneration, may enable in silico simulation using a super computer of the components and functions of nucleolus that shift dynamically under physiological and pathological conditions.

Another issue is the relationship between proteasome-dependent protein degradation and nucleolar protein dispersion in the nucleus under osmotic stress conditions and in response to polyQ protein expression. Our results showed that fibrillarin is dispersed but was still expressed, with the fibrillarin signals being weaker in response to polyQ protein expression than to osmotic stress. Hyperosmotic stress has been reported to induce nuclear foci, including proteasomal proteins and VCP[86]. These findings suggested that LLPS foci function in protein degradation, based on the correlation between proteasome inhibitor treatment and the number/size of foci and the effect of ubiquitin inhibitor pre-treatment and formation of foci[86]. However, there is no evidence showing protein degradation at LLPS foci. Conversely, LLPS foci may be a warehouse for proteasome proteins in which they do not function, such that some proteins may not be digested by proteasomes under osmotic stress conditions but that mutant proteins are degraded. Alternatively, these results may be due to differences in duration of the two experiments. Nucleolar proteins were evaluated for 24 h, whereas cells were subjected to osmotic shock for only 20 min, during which time the proteasome system was activated[86]. By contrast, the effects of polyQ protein expression were observed for 72 h after transfection. Even if the levels of proteasome activation were similar, proteins would be more degraded in the polyQ expression experiments. Further studies are needed to determine the mechanism underlying differences in the rates of nucleolar protein degradation.

Consistent with findings showing that PQBP5 functions to anchor the nucleolus to the rRNA transcription site, high-speed AFM analysis suggested that PQBP5 interacts with cleaved rRNA, and droplet assays indicated that rRNA promotes the self-assembly of PQBP5 and its co-assembly with other nucleolar proteins. The apoptosis-antagonizing transcription factor AATF/Che-1/TRB[87–89], neuroguidin (NGDN), and PQBP5/NOL10 have been reported to form a complex for generating the 40S ribosomal subunit[90]. Therefore, PQBP5-rRNA interaction could be the basis for localizing nucleoli at NORs. Future analysis may fill a gap in knowledge of the general concept of 40S ribosome generation[69].

The mechanisms by which PQBP5 interacts with different aggregation states of Atxn1 and htt should also be analyzed. Solid-state NMR studies of htt fibrils indicate that polyQ proteins form a dense core, often excluding polyQ binding proteins such as anti-polyQ

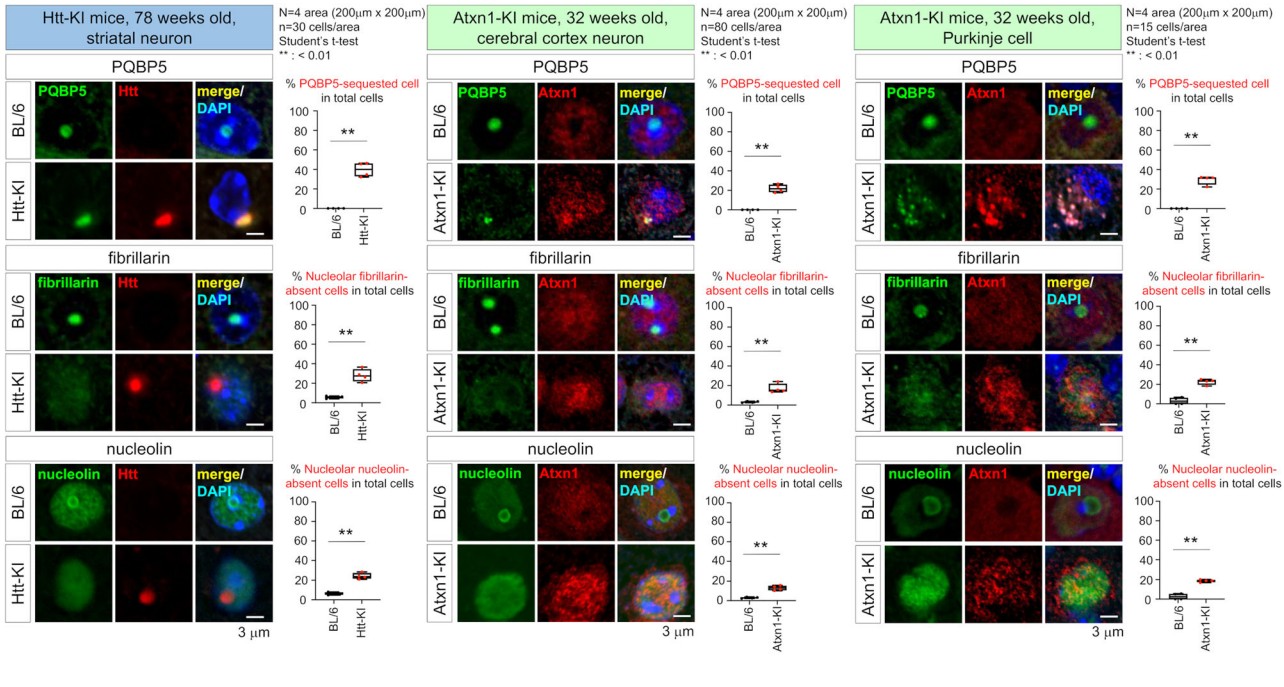

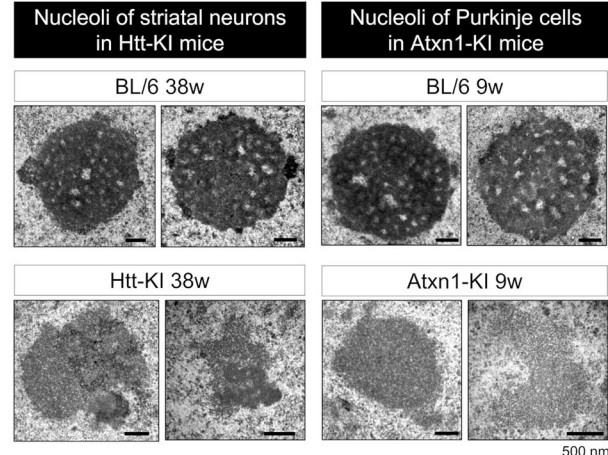

**Fig. 7 | Effects of neurodegenerative proteins on PQBP5 and nucleolus decomposition. a** Immunohistochemistry of PQBP5 together with normal or mutant polyglutamine disease proteins. PQBP5 was sequestered to the inclusion bodies formed by mutant Htt in the striatal neurons of mutant Htt-KI mice (upper left panels) and in the inclusion bodies formed by mutant Atxn1 in the cortical neurons of mutant Atxn1-KI mice (upper middle panels). In some Purkinje cells of mutant Atxn1-KI mice, PQBP5 was sequestered to cytoplasmic inclusions (upper right panels). In all cases, fibrillarin and nucleolin lost their nucleolar assembly activity. Fibrillarin disappeared from the nucleoli, and nucleolin was dispersed in the nucleus. The right graphs in the left and middle columns show quantification of PQBP5-sequestered cells (upper graphs), cells lacking nucleolar fibrillarin (middle graphs), and cells lacking nucleolar nucleolin (lower graphs). The box plot shows median, 25–75th percentile, and whiskers representing data outside the 25–75th

percentile range. Two-sided Student's *t*-test was used for statistical examination. *P*-values are as follows. % PQBP5-sequestered cells: 0.0017, % fibrillarin-absent cells: 0.0053, % nucleolin-absent cells: 0.0003 in striatal neurons of Htt-KI mice at 78 weeks. % PQBP5-sequestered cells: 0.0018, % fibrillarin-absent cells: 0.0095, % nucleolin-absent cells: 0.0026 in cerebral cortex neurons of Atxn1-KI mice at 32 weeks. % PQBP5-sequestered cells: 0.0011, % fibrillarin-absent cells: 0.0002, % nucleolin-absent cells: 0.0006 in Purkinje cells of Atxn1-KI mice at 32 weeks. **b** EM analysis of nucleoli in Purkinje cells of mutant Atxn1-KI mice and in striatal neurons of mutant Htt-KI mice. The right bar graph shows the percentage of abnormal cells having morphological features of abnormal nucleoli (irregular shape, low electron density, loss of FC/DFC) in four mouse genotypes. Data are presented as mean values ± SEM. Tukey's HSD test was used for multiple comparisons. *P* < 0.0001 in comparisons by GraphPad Prism.

antibodies[91]. Thus, PQBP5 may initially interact with soluble forms of polyQ proteins before being pulled into larger inclusions. This is consistent with the hypothesis that LLPS-based droplets of pure mutant protein mature to aggregates of fibrils[92], as well as with results showing that LLPS droplets composed of multiple normal proteins, such as PQBP1 and cGAS, remain soluble and are able to activate the cGAS-STING signaling pathway[30,93]. Studies are needed to determine the mechanism by which aggregation inhibitors that prevent inclusion formation affect the distribution of PQBP5 and other nucleolar

proteins in polyQ disease pathology. Such an approach might provide novel ideas for future therapeutics.

## Methods
### Plasmid construction
To construct pCMV-3Tag1A plasmids expressing full-length human PQBP5 and PQBP5 deletion mutants, total RNA was isolated from HEK293 cells (TaKaRa, Kusatsu, Shiga, Japan), and reverse transcribed to cDNA. Full-length PQBP5, encoding amino acids 1-688, was

amplified using the primers 5′-CCCGAATTCATGCAGGTCTCCAGC-3′ (forward) and 5′-GGGGGTACCTCAATGAAACGACCGTC-3′ (reverse). In addition, sequences corresponding to amino acids 82-688, 205-688, 341-688, and 482-688 were amplified using the forward primers 5′-CCCGAATTCATGACCTATCAATTAT-3′, 5′-CCCGAATTCATGCCAAGAA CTCGAA-3′, 5′-CCCGAATTCATGATTCCAGTTTTGG-3′, and 5′-CCCGA ATTCATGGATGATCGATTTA-3′, respectively, and the reverse primer 5′-GGGGGTACCTCAATGAAACGACCGTC-3′. The sequence corresponding to amino acids 1-481 was amplified using the primers 5′-CCCGAATTCATGGATGATCGATTTA-3′ (forward) and 5′-GGGGGTAC CTCAATGAAACGACCGTC-3′ (reverse), and the sequences corresponding to amino acids 1-340, 1-204, and 1-81 were amplified using the forward primer 5′-CCCGAATTCATGCAGGTCTCCAGC-3′ and the reverse primers 5′-ATGCGGTACCTCAGTAATAGATGCCCATCTTGGG-3′, 5′-ATGCGGTACCTCAGTCCCAGCACTCCACTCTA-3′, and 5′-ATGCG GTACCTCAGTCATAACATCGAACCCGAG-3′, respectively. After digestion with EcoRI and KpnI, each of these sequences was subcloned into pCMV-3Tag-1A. In addition, the full-length human PQBP5 cDNA, amplified as above, was subcloned into the plasmid pEGFP-N1.

To construct pET-28a plasmids expressing full-length human PQBP5 and PQBP5 deletion mutants, sequences corresponding to amino acids 1-688, 82-688. 205-688, 341-688, and 482-688 were amplified from HEK293 cDNA, prepared as above using the forward primers 5′-ATGCGAATTCATGCAGGTCTCCAGCC-3′, 5′-CCCGAATT-CATGCAGGTCTCCAGCC-3′, 5′-CCCGAATTCATGCCAAGAACTCGAA-3′ 5′-CCCGAATTAATGATTCCAGTTTTGG-3′, and 5′-CCCGAATTCATG-GATGATCGATTTA-3′, respectively, and the reverse primer 5′-CATG CGGCCGCTCAATGAAACGACCGTC-3′, The sequence corresponding to amino acids 1-481 was amplified using the forward primer 5′-CCC GAATTCATGCAGGTCTCCAGC-3′ and the reverse primer 5′-GGGGCG GCCGCTCAGGTGAGAATATTA-3′. Each of these sequences was digested with EcoRI and NotI and subcloned into pET-28a.

A pET-28a plasmid expressing human fibrillarin was constructed by amplifying a fibrillarin sequence from HEK293 cDNA using the primers 5′-CATGCGGCCGCTCAGTTCTTCACCTTGGGGGGT-3′ (forward) and 5′-ATGCGAATTCATGAAGCCAGGATTCAGTCCCC-3′ (reverse). A pET-28a plasmid expressing human nucleolin was constructed by amplifying a nucleolin sequence from HEK293 cDNA using the primers 5′-CATGAATTCCTATTCAAACTTCGTCTTCTT-3′ (forward) and 5′-ATGCCATATGGTGAAGCTCGCGAAGGCA-3′ (reverse). Each of these sequences was digested with EcoRI and NdeI and subcloned into pET-28a.

The pGEX-6P-1ATXN1-33Q/86Q plasmid was constructed by digesting the plasmid pCMV-myc-AT1-33Q/86Q with EcoRI and NotI and subcloning the resulting Atxn1-33Q/86Q cDNA into pGEX-6P-1[94]. The plasmid ATXN1-33Q-86Q-DsRed was constructed by subcloning full-length ATXN1 cDNA into the plasmid pDsRed-monomer-C1 (Clontech, Mountain View, CA, USA) between the XhoI and EcoRI sites. The Htt-17Q/103Q-DsRed plasmid was constructed by amplifying Htt-exon 1 cDNA from pEGFP-Htt-17Q/88Q using the primers 5′-CAT-GAATTCTATGGCGACCCTGGAAAAG-3′ (forward) and 5′-CATG-GATCCTCACGGTCGGTGCAGCGGCT-3′ (reverse) and subcloning the sequence between the EcoRI and BamHI sites of pDsRed-monomer-C1[94].

### Sample preparation for high-speed AFM and Droplet assays
Rosetta™ 2(DE3) Singles™ Competent *E. coli* cells (#71400, Sigma Aldrich, St. Louis, MO, USA) were transformed individually with pET-28a-human fibrillarin, pET-28a-human nucleolin, and pET-28a-human PQBP5. The bacteria were grown to an $OD_{600}$ of 0.5 and incubated with 1 mM IPTG for 18 hours at 16 °C to induce expression of the fusion protein, followed by the centrifugation of 800 ml of each bacterial culture at 3, 000 × *g* for 20 min at 4 °C. To purify His-tag fusion proteins, the pellets were resuspended in lysis buffer (25 mM Tris-HCl, 500 mM NaCl, 10% (vol/vol) glycerol, 1% (vol/vol) Triton X-100, 10 mM

imidazole, final pH 8.0) containing 0.1 mg/ml lysozyme and protease inhibitor cocktail (#539134, Merck Millipore, Burlington, MA, USA). After 30 min on ice, the cells were sonicated using an Ultrasonic Dispersion Machine UH-50 at level 6 for 10 min and centrifuged at 16,000 × *g* for 20 min at 4 °C. His$_6$-tagged proteins were purified on Ni-NTA agarose (#30210, Qiagen, Hilden, Germany), which was washed with wash buffer (25 mM Tris-HCl, 500 mM NaCl, 10% glycerol, 1% Triton X-100, 20 mM imidazole, final pH 8.0). The proteins were eluted with elution buffer (25 mM Tris-HCl, 500 mM NaCl, 10% glycerol, 1% Triton X-100, 250 mM imidazole, final pH 8.0) and stored at −80 °C. Protein samples were quantified by CBB staining and stored at −80 °C before use. Although RNase inhibitor was not used, the protein samples included a smear of small RNAs on agar gel electrophoresis. RT-PCR for three or four control genes of *E. coli*[95] showed no evidence of mRNA or tRNA contamination.

### High-speed AFM
A glass sample stage (2 mm in diameter and 2 mm in height) with a thin mica disk 1 mm in diameter and 0.05 mm thick, bonded with epoxy, was attached to the top of a Z scanner with drops of nail polish. New mica surfaces were prepared by removing the top layer of mica with adhesive tape. One drop (2 μe of protein (about 3 nM) diluted with dilution buffer (20 mM Tris-HCl (pH 7.5), 500 mM NaCl) was attached to the mica surface. After incubation for 3 min, the mica surface was rinsed with 20 μi of observation buffer to remove the suspended sample. The sample stage was immersed in a liquid cell containing about 60 μe of the same observation buffer (20 mM Tris-HCl, pH 7.5). HS-AFM observation was performed in tapping mode using a laboratory-built device[96,97]. The custom-made short cantilever (BL-AC7DS-KU4, Olympus, Tokyo, Japan) had a resonant frequency of about 1 MHz and a quality factor of about 2 in water, as well as a spring constant of 0.1–0.15 N m$^{-1}$. The free vibration amplitude of the cantilever, $A_0$, was set to 1–2 nm, and the set point amplitude, As, to about $0.9 × A_0$, with the average loss of vibration energy of the cantilever per tap adjusted to 1–3 k$_B$T.

### Protein preparation for SPR analysis
Full-length and deletion mutant PQBP5 and polyQ proteins were expressed in *E. coli* cells (#71400, Sigma Aldrich) by transformation of pET-28a-human PQBP5 and pET-28a-human PQBP5 deletion mutants or by transformation of pGEX-6P-1-AT1-33Q; pGEX-6P-1-AT1-86Q; pGEX-3X-Htt-17Q; and pGEX-3X-Htt-103Q[98], as described above. *E. coli* cells were collected by centrifugation, and His-tag PQBP5 fusion proteins were purified as described above. For GST-polyQ fusion proteins, the pellets were resuspended in lysis buffer (50 mM NaH$_2$PO$_4$, 300 mM NaCl, 1 mM DTT, 0.1% Triton X-100) containing 0.1 mg/ml lysozyme and protease inhibitor cocktail (#539134, Merck Millipore). After 30 min on ice, the cells were sonicated at level 6 for 10 min and centrifuged at 16,000 × *g* for 20 min at 4 °C. GST fusion proteins were added to glutathione Sepharose 4B (#17-0756-05, GE Healthcare, Chicago, IL, USA). After washing with wash buffer (50 mM NaH$_2$PO$_4$, 300 mM NaCl, 1 mM DTT), the proteins were eluted with elution buffer (50 mM Tris-HCl (pH 8.0), 1% Triton X-100, 20 mM glutathione).

### SPR analysis
SPR analysis was performed using a Biacore T100 instrument (Cytiva, Marlborough, MA, USA) at 25 °C. Interactions among the proteins PQBP5, fibrillarin, and nucleolin were assessed by immobilizing 5 ng/ml of each protein dissolved in 10 mM sodium acetate on a CM5 sensor chip until the immobilized proteins had gained 200 resonance units (RU). Multiple concentrations of analytes (0, 5, 10, 20, 30, 40, and 50 nM) were injected for 180 s at a rate of 10 μl/min. Sample binding and dissociation were assessed in HBS-EP$^+$ buffer for 300 s. Except for the binding of PQBP5 as analyte and nucleolin as immobilized protein, all binding experiments were performed in His-tagged protein elution

buffer without imidazole (25 mM Tris-HCl, 500 mM NaCl, 10% glycerol, 1% Triton X-100, final pH 8.0). The sensor chip was regenerated by injection of 10 mM glycine-HCl (pH 2.1) for 60 sec at a rate of 30 µl/min to perform kinetic analysis.

To assess the binding of polyglutamine disease proteins (GST-Atxn1-33Q, GST-Atxn1-86Q) to PQBP5 or its deletion mutants, 5 ng/ml of the Atxn1 proteins in 10 mM sodium acetate were immobilized on a CM5 sensor chip until the immobilized proteins had gained 11,000 RU. Multiple concentrations of analytes (0, 12.5, 25, 50, 100, and 200 nM) were injected for 180 s at a rate of 10 µl/min. Sample binding and dissociation were assessed in HBS-EP+ buffer for 300 s. The sensor chip was regenerated as described above.

To determine the binding of polyglutamine disease proteins (GST-Htt-20Q, GST-Htt-110Q) to PQBP5 or its deletion mutants, 5 ng/ml Htt proteins in 10 mM sodium acetate solution were immobilized to 1100 RU on a CM5 sensor chip. Multiple concentrations of analytes (0, 12.5, 25, 50, 100, and 200 nM) were injected for 180 s at a rate of 20 µl/min. Sample binding and dissociation were assessed in HBS-EP+ buffer for 300 s, and the sensor chip was regenerated as described above.

### Immunoprecipitation

HeLa cells (RCB0007), a kind gift from Naoyuki Kataoka that was purchased from RIKEN BRC Cell Bank (Tsukuba, Japan), were cultured with Dulbecco's Modified Eagle's Medium–high glucose (D5796 medium, Sigma Aldrich, St. Louis, MO, USA) with 10% Fetal Bovine Serum (#10270106, Gibco, MA, USA), transfected with the plasmids of pCMV-3Tag1A-Full-length PQBP5 or PQBP5 deletion mutants and pCMV-myc-AT1-33Q/86Q or pCMV-myc-Htt-Exon1-17Q/103Q, and harvested 48 h later. U2OS cells, a generous gift from Professor Yoshio Miki (Tokyo Medical and Dental University) that was purchased from ATCC (Manassas, VA, USA), were cultured in Dulbecco's Modified Eagle's Medium–high glucose (D5796 medium, Sigma Aldrich, St. Louis, MO, USA) with 10% Fetal Bovine Serum (#10270106, Gibco, MA, USA), transfected with the plasmids of pCMV-3Tag1A-Full-length PQBP5, pCMV-myc-AT1-33Q/86Q or pCMV-myc-Htt-Exon1-17Q/103Q, and harvested 48 h later.

The cells were lysed with TNE buffer (10 mM Tris-HCl (pH 7.5), 150 mM NaCl, 1 mM EDTA, 1% Nonidet P-40) containing protease inhibitor cocktail (#539134, Merck Millipore). The lysates were rotated for 60 min at 4 °C and centrifuged at 12,000 × g for 1 min at 4 °C. Each supernatant was incubated with a 50% slurry of Protein-G Sepharose beads (17061801, GE Healthcare, Chicago, IL, USA) for 2 h at 4 °C, followed by centrifugation at 2000 × g for 2 min at 4 °C. The supernatants were incubated with 2 µp antibody for 5 h at 4 °C with rotation, followed by the addition of 40 µl Protein-G Sepharose and rotation for 2 h at 4 °C. The beads were washed three times with TNE buffer, and 30 µl sample buffer (125 mM Tris-HCl (pH 6.8), 4% (w/v) SDS, 5% (v/v) 2-mercaptoethanol, 10% (v/v) glycerol, and 0.0025% (w/v) bromophenol blue) were added to each sample. The samples were boiled at 95 °C for 10 min, followed by SDS-PAGE and transfer to Immobilon-P polyvinylidene difluoride membranes (Merck Millipore). The membranes were incubated with various primary antibodies, including rabbit anti-FLAG antibody (#F7425, Sigma); mouse anti-Myc antibody (#M047-3, MBL, Aichi, Japan); rabbit anti-fibrillarin antibody (1:1000 #ab166630, Abcam, Cambridge, UK); rabbit anti-nol10 antibody (1:5000 #ab181161 Abcam); and mouse anti-nucleolin antibody (1:5000 #ab13541 Abcam).

### Super-resolution microscopy

HeLa cells were fixed in 4% formaldehyde for 15 min at RT, permeated with 0.1% Triton X-100, and blocked by incubation in blocking buffer (50 mM Tris-HCl pH 6.8, 150 mM NaCl, and 0.1% Tween-20 in MQ) containing 1 mg/ml BSA and 300 mM glycine for 60 min at RT. The samples were incubated with mouse anti-fibrillarin (1:100, #ab4566, Abcam), mouse anti-NPM1 (1:5000, #ab10530, Abcam), rat anti-PES1 antibody (1:2000, #252849, Abcam), or rabbit anti-NOL10/PQBP5

(1:150 #ab181161 Abcam) antibody for 60 min at RT, followed by incubation for 60 min at RT with the secondary antibodies Alexa Fluor 555-conjugated anti-mouse IgG (1:1000, #A31570, Molecular Probes, Eugene, OR, USA), donkey anti-mouse IgG (H + L) Highly Cross-Adsorbed Alexa Fluor 647 (1:1000, #31571, Thermo Fisher Scientific, Waltham, MA, USA), donkey anti-rat IgG (H + L) Highly Cross-Adsorbed Alexa Fluor 488 (1:1000, #21208, Thermo Fisher Scientific, Alexa Fluor 488-conjugated anti-rabbit IgG (1:1000, #A21206, Molecular Probes) or donkey anti-rabbit IgG (H + L) Highly Cross-Adsorbed Alexa Fluor 568 (1:1000, #10042, Thermo Fisher Scientific). For nucleolin staining, the samples were incubated with 10% normal goat serum for 60 min, followed by incubation for 60 min at RT with mouse anti-nucleolin antibody (1:2000 #ab13541 Abcam) that had been labeled with the Zenon™ Alexa Fluor 647 Mouse IgG1 labeling kit (#z25008, Thermo Fisher Scientific). The samples were fixed with 4% formaldehyde for 15 min at RT, and the 3D images were acquired by SRM, using an Elyra 7 (Carl Zeiss Co., Ltd., Oberkochen, Germany), LSM 980 with Airyscan2 (Carl Zeiss Co., Ltd), or IX83 (Olympus, Tokyo, Japan) microscope equipped with a CSU-W1 SoRa Spinning Disk Confocal (Yokogawa Electric Corporation, Tokyo, Japan), an ORCA Flash4.0 V3 digital CMOS camera (Hamamatsu, Shizuoka, Japan), and a UPLXAPO60XO objective lens (Olympus, Tokyo, Japan).

### Correlative light and electron microscopy

HeLa cells fixed in 4% formaldehyde for 15 min at RT were incubated with antibodies to nucleolin; nol10; and fibrillarin, as described above for immunocytochemistry. The cells were visualized using an LSM 980 with an Airyscan2 super-resolution microscope (Carl Zeiss, Co., Ltd), fixed in 2.5% glutaraldehyde in 0.1 M phosphate buffer at 4 °C for 2 h, and incubated with 1% osmium tetroxide at 4 °C for 2 h. The cells were dehydrated with a graded series of ethanol, and embedded in epon at 60 °C for 48 h and at 120 °C for 24 h. Ultrathin sections (80 nm) were prepared with an ultramicrotome (US6, Leica, Wetzlar, Germany), incubated with uranyl acetate and lead citrate, and viewed under a scanning electron microscope (JSM-7900F, JEOL, Tokyo, Japan).

### shRNA-expressing plasmids

The shRNA-expressing plasmids for PQBP5 (SH837805), nucleolin (SH843928), fibrillarin (SH827603) and their scramble shRNA controls were generated by Vigene Biosciences (Rockville, MD, USA). Briefly, the shRNA sequences for PQBP5 (NM_024894), nucleolin (NM_005381), and fibrillarin (NM_001436) consisted of 5′-GAAGTCCGTTCATTTCCAGGATTCTTCAAGAGAGAATCCTGGAA ATGAACGGACTTTTTTT-3′, 5′- GAAGAAGAGGACCAACATCATTCC TTCAAGAGAGGAATGATGTTGGTCCTCTTCTTTTTTTT-3′ and 5′-GA AACAGAAAGCAGCTCCTGAAGCTTCAAGAGAGCTTCAGGAGCTGC TTTCTGTTTTTTTTT-3′, respectively. Each of these sequences and a scramble control shRNA sequence were cloned into the plasmid pAV-U6-GFP between the BamHI and HindIII sites at the 3′ terminal of the human U6 promoter.

### Western blot

Forty-eight hours after transfection of shRNA plasmids, cells were harvested and incubated for 15 min at 4 °C in lysis buffer (10 mM Tris-HCl (pH 7.5), 150 mM NaCl, 1 mM EDTA, 1% NP-40) with protease inhibitor cocktail (#539134, 1:200 dilution, Calbiochem, San Diego, CA, USA). After centrifugation at 10,000 × g for 10 min, the supernatants were mixed with an equal volume of sample buffer (125 mM Tris-HCl (pH 6.8), 4% (w/v) SDS, 5% (v/v) 2-mercaptoethanol, 10% (v/v) glycerol, and 0.0025% (w/v) bromophenol blue), separated by SDS-PAGE, and transferred to Immobilon-P polyvinylidene difluoride membranes (Merck Millipore) by the semi-dry method. The membranes were blocked by incubation in 5% milk in TBST (10 mM Tris/HCl (pH 8.0), 150 mM NaCl, 0.05% Tween-20), and incubated overnight at 4 °C with rabbit anti-fibrillarin (1:1000, #ab166630); mouse anti-C23 (1:5000,

#sc-8031, Santa Cruz Biotechnology, Dallas, TX, USA); rabbit anti-nol10 (1:5000 #ab181161, Abcam); rabbit anti-GFP (1:6000, #8334, Santa Cruz Biotechnology), or mouse anti-GAPDH (1:6000, #MAB374, Merck Millipore) primary antibodies, each diluted in Can Get Signal solution (Toyobo, Osaka, Japan). The membranes were subsequently incubated with HRP-linked anti-rabbit IgG (1:3000, #NA934, GE Healthcare) or HRP-linked anti-mouse IgG (1:3000, #NA931, GE Healthcare) secondary antibodies for 1 h at room temperature. Proteins were detected with ECL Prime Western Blotting Detection Reagent (RPN2232, GE Healthcare) and a luminescent image analyzer (ImageQuant LAS 500, GE Healthcare). Uncropped and unprocessed scans of the blots are included in Source Data file.

## Immunoelectron microscopy

Hela cells were transfected with 2.5 µg each of the plasmids pAV-U6-GFP-nucleolin shRNA (SH843928), pAV-U6-GFP-fibrillarin shRNA (SH827603), and pAV-U6-GFP-PQBP5 shRNA (SH837805), each pre-pared by Vigene Biosciences, using Lipofectamine 2000 transfection reagent (Thermo Fisher Scientific, #11668019). After 48 h, the cells were fixed with fixation buffer (4% paraformaldehyde, 0.1% glutar-aldehyde in 0.1 M phosphate buffer) for 15 min at 25 °C and incubated in blocking buffer (50 mM $NH_4Cl$, 0.1% Saponin, 1% BSA) for 1 hour at 25 °C. The cells were stained with rabbit anti-GFP antibody (1:10000, #sc-8334, Santa Cruz Biotechnology) for 30 min at 25 °C, and incubated with Nanogold®-IgG goat anti-rabbit IgG (H + L) (1:200, #2003, Nanoprobes, Yaphank, NY, USA) for 2 h at 25 °C. Nanogold signals were enhanced with GoldEnhance™ EM (#2113, Nanoprobes) for 5 min at 25 °C. After fixation in 1% glutaraldehyde in 0.2 M HEPES, the sections were post-fixed with 1% OsO4 for 30 min at 4 °C and dehydrated through a graded ethanol series. The samples were incubated twice with propylene oxide for 15 min each, once with a 1:1 mixture of pro-pylene oxide and epon for 4 h at 25 °C, once with a 1:3 mixture of propylene oxide and epon overnight at 25 °C, and embedded in pure epon for 3 days. Ultrathin sections (80 nm) were prepared with a ultramicrotome (UC7, Leica, Wetzlar, Germany) and stained with ura-nyl acetate and lead citrate. The sections were observed using a transmission electron microscope (model 1400 plus, JEOL Ltd).

## Osmotic stress and Immunocytochemistry

HeLa cells were incubated in hypotonic buffer (D1152 medium, Sigma), isotonic buffer (D1152 medium with 44 mM $NaHCO_3$), hypertonic buffer (D1152 medium with 44 mM $NaHCO_3$ and 0.2 M sorbitol (#191-14735, Fujifilm, Osaka, Japan), or super-hypertonic buffer (D1152 medium with 44 mM $NaHCO_3$ and 0.4 M sorbitol (#191-14735, Fujifilm). The cells were subsequently incubated in isotonic buffer (D1152 med-ium with 44 mM $NaHCO_3$) for 0, 10, 30, 180, 720, or 1440 min at 37 °C. Following cell fixation in 4% formaldehyde for 15 min at RT, the cells were washed three times in PBS, permeated with 0.1% Triton X-100, and blocked with blocking buffer (50 mM Tris-HCl pH 6.8, 150 mM NaCl, and 0.1% Tween-20 in MQ) containing 1 mg/ml BSA and 300 mM glycine for 60 min at RT. The samples were subsequently incubated with both mouse anti-fibrillarin (1:100, #ab4566, Abcam) or rabbit anti-nol10 (1:150 #ab181161, Abcam) for 60 min at RT followed by incuba-tion with Alexa Fluor 488-conjugated anti-rabbit IgG (1:1000, #A21206, Molecular Probes) and Alexa Fluor 555-conjugated anti-mouse IgG (1:1000, #A31570, Molecular Probes) secondary antibodies for 60 min at RT. For nucleolin staining, the cells were incubated with 10% normal goat serum for 60 min, followed by incubation at RT for 60 min with mouse anti-nucleolin antibody (1:2000, #ab13541, Abcam) that had been labeled with a Zenon™ Alexa Fluor 647 Mouse IgG₁ labeling kit (#z25008, Thermo Fisher Scientific) and fixation with 4% for-maldehyde for 15 min at RT.

For the immunocytochemistry by super-resolution microscopy and correlative light and electron microscopy, HeLa cells were cul-tured with Dulbecco's Modified Eagle's Medium – high glucose (D5796 medium, Sigma Aldrich, St. Louis, MO, USA) with 10% Fetal Bovine Serum (#10270106, Gibco, MA, USA) for 24 h at 37 °C before fixation. For knockdown experiment, HeLa cells were transfected with the shRNA-expressing plasmids for PQBP5 (SH837805), nucleolin (SH843928), fibrillarin (SH827603) and their scramble shRNA controls. Forty-eight hours after transfection, cells were fixed with 4% formaldehyde.

For immunocytochemistry by confocal microscopy, U2OS cells were transfected with the plasmids of ATXN1-33Q/86Q-DsRed or Htt-17Q/103Q-DsRed. After 72 h, cells were fixed with 4% formaldehyde.

## Analysis of 47 S rRNA expression after PQBP5 knockdown

HeLa cells were transfected with 3 µg of pAV-U6-GFP-scrambled shRNA or pAV-U6-GFP-PQBP5 shRNA plasmid using Lipofectamine 2000 transfection reagent (Thermo Fisher Scientific, #11668019). The cells were harvested at 48 or 72 h after transfection, and divided for western blot, RT-PCR and quantitative RT-PCR analyses. For western blot ana-lysis, cells were lysed with TNE buffer (10 mM Tris-HCl (pH 7.5), 150 mM NaCl, 1 mM EDTA, 1% NP-40) containing protease inhibitor cocktail (#539134, 1:200 dilution, Calbiochem, San Diego, CA, USA) for 15 min at 4 °C. After centrifugation at 10,000 g for 10 min), the supernatants were mixed with an equal volume of sample buffer (125 mM Tris-HCl (pH 6.8), 4% (w/v) SDS, 5% (v/v) 2-mercaptoethanol, 10% (v/v) glycerol, and 0.0025% (w/v) bromophenol blue), separated by SDS-PAGE, and transferred to Immobilon-P polyvinylidene difluor-ide membranes (Merck Millipore) by the semi-dry method. The membranes were incubated with rabbit anti-Nol10 antibody (1:5000 #ab181161, Abcam) overnight at 4 °C, followed by incubation with HRP-linked anti-rabbit IgG (1:3000, #NA934, GE Healthcare) for 1 h at room temperature. For RT-PCR analysis, total RNA was isolated from the samples with RNeasy mini kits (74106, Qiagen, Limburg, Netherlands). During RNA isolation, DNA was digested on-column using DNase I to eliminate genomic DNA contamination. Reverse transcription was performed using the SuperScript VILO cDNA Synthesis kit (11754-250, Invitrogen, Carlsbad, CA, USA), and PCR was performed with the T100 Thermal Cycler (Bio-Rad, Hercules, CA, USA) using TaKaRa Ex Taq (#RR001A, TaKaRa, Shiga, Japan). The PCR amplification protocol consisted of an initial denaturation for 2 min at 95 °C, followed by 20 cycles of denaturation for 25 s at 95 °C, annealing for 30 sec at 55 °C, and extension for 60 s at 72 °C. After mixing with loading buffer (#9157, TaKaRa), the samples were separated by 2% agarose gel elec-trophoresis in 1 × TAE buffer (0.04 M Tris–acetate, 1 mM EDTA). Band intensity on western blots and agarose gels were analyzed using Image J software. For quantitative RT-PCR, analyses were performed with the 7500 Real-Time PCR System (Applied Biosystems, Foster City, CA, USA) using the Thunderbird SYBR Green (QPS-201, TOYOBO, Osaka, Japan) and assessed by the standard curve method. The PCR condi-tions for amplification were 95 °C for 10 min for enzyme activation, 95 °C for 15 s for denaturation, and 60 °C for 1 min for extension (40 cycles).

The PCR primer sequences were:
Human 47S rRNA:[99]
forward primer: 5′-GCTGACACGCTGTCCTCTG-3′
reverse primer: 5′-ACGCGCGAGAGAACAGCAG-3
Human GAPDH:[100]
forward primer: 5′- GGTGGTCTCCTCTGACTTCAACA-3′
reverse primer: 5′- GTTGCTGTAGCCAAATTCGTTGT-3

## Ribosomal RNA preparation for Droplet assay

Total RNA was isolated from HeLa cells with RNeasy mini kits (74106, Qiagen), including on-column digestion of DNA with DNase I to elim-inate genomic DNA contamination. The total RNAs were hybridized with the single-stranded DNA probes specific for ribosomal RNAs (NEB #7405, New England BioLabs, Ipswich, MA, USA); residual non-hybridized single-strand RNAs were degraded using RNase A

(#EN0531, Thermo Fisher Scientific), and the hybridized DNA probes were degraded by NEBNext DNase I (NEB #7405, New England Bio-Labs). Ribosomal RNAs were recovered by NEBNext RNA Sample purification beads (NEB #7405, New England BioLabs) and stored at −80 °C.

## Droplet assay

AlexaFluor 488, AlexaFluor 647, and AlexaFluor 555 were conjugated to PQBP5, fibrillarin, and nucleolin proteins, respectively, using the #A30006, #A30009, and #A30007 AlexaFluor Labeling kits (Thermo Fisher Scientific), respectively. The reaction buffer was removed by dialysis against the low osmolarity buffer (20 mM Tris-HCl, pH 7.5, 25 mM NaCl, and 1 mM DTT) for droplet formation. All protein concentrations were adjusted to 0.25 μM in hypotonic (20 mM Tris-HCl, pH 7.5, 125 mM NaCl, 1 mM DTT); isotonic (20 mM Tris-HCl, pH 7.5, 125 mM NaCl, 40 mM NaHCO$_3$, 1 mM DTT); hypertonic (20 mM Tris-HCl, pH 7.5, 125 mM NaCl, 40 mM NaHCO$_3$, 0.2 M sorbitol, 1 mM DTT); and superhypertonic (20 mM Tris, pH 7.5, 125 mM NaCl, 40 mM NaHCO$_3$, 0.4 M sorbitol, 1 mM DTT) solutions. Droplet formation on silicone wells (#GBL665108-25EA, Grace BioLabs, Bend, OR, USA) in the absence or presence of 2.5 μg/ml ribosomal RNA was assessed by incubation for 0, 3, 6, and 9 hours at 25 °C, and observed by confocal microscopy (FV1200, Olympus).

## Thermal and osmolar shift assay

PQBP5, nucleolin, and fibrillarin His-tag fusion proteins were generated as described above, adjusted to 0.3 μM, and dialyzed against to 20 mM Tris-HCl, pH 7.5, 25 mM NaCl, 1 mM DTT using Slide-A-Lyzer MINI Dialysis Units (#69570, Thermo Fisher Scientific). Each well of a PCR plate (MicroAmp™ Fast Optical 96-Well Reaction Plate, #4346907, Live Technologies, Carlsbad, CA, USA) contained 10 μl of a fusion protein, 2.5 μl of Protein Thermal Shift dye (#4462263, Thermo Fisher Scientific) and 7.5 μl of osmotic buffer, resulting in a final volume of 20 μl. Four types of osmotic buffer were used: hypotonic (246.5 mM NaCl); isotonic (353 mM NaCl); hypertonic (620 mM NaCl); and superhypertonic (886.5 mM NaCl). The plates were sealed with MicroAmp Optical Adhesive Film, and the temperature was increased from 25 °C to 99 °C at a rate of 0.4 °C per minute, during which time the fluorescence in each well was monitored. The results were analyzed using Protein Thermal Shift™ Software v1.4 (#4466037, Thermo Fisher Scientific).

## Mice

Mutant *Ataxin-1* knock-in mice (*Sca1*$^{154Q/2Q}$ mice) were a generous gift from Prof. Huda Y. Zoghbi (Baylor College of Medicine, Houston, TX, USA)[101], and mutant Huntingtin knock-in mice (*Hdh*$^{Q111}$ mice) were a generous gift from Prof. Marcy MacDonald (Massachusetts General Hospital, Harvard Medical School, Boston, MA, USA)[102]. C57BL/6J (BL/6) mice purchased from Sankyo Labo Service Corporation (Tokyo, Japan) were used for breeding. Non-transgenic sibling mice were used as controls. All animal experiments were performed in accordance with Animal Research: Reporting in vivo Experiments (ARRIVE) guidelines and were approved by the Committees on Gene Recombination Experiments and Animal Experiments of Tokyo Medical and Dental University (G2018-082C3 and A2021-211A).

## Immunohistochemistry

Mouse brains were fixed in 4% paraformaldehyde for 16 h, embedded in paraffin, and sectioned at a thickness of 5 μ5. The sections were de-paraffinized in xylene and re-hydrated, followed by antigen retrieval by microwaving in 0.01 M citrate buffer (pH 6.0) at 120 °C for 15 min. After permeation with PBS containing 0.5% Triton X-100, the sections were incubated with blocking solution (10% normal donkey serum in PBS) for 60 min at room temperature, following by incubation for 24 hours at 25 °C with mouse anti-Htt (1:100, MAB5374, Sigma Aldrich), mouse anti-Atxn1 (1:100, MAB5374, Sigma Aldrich), mouse anti-fibrillarin (1:100, #ab4566, Abcam), rabbit anti-nol10 (1:5000, #ab181161, Abcam), and mouse anti-nucleolin (1:5000, #ab13541, Abcam) primary antibodies. After washing three times in PBS at room temperature, the sections were incubated at room temperature for 1 h with Alexa Fluor 488-conjugated anti-rabbit IgG (1:1000, A21206, Molecular Probes) and Alexa Fluor 555-conjugated anti-mouse IgG (1:1000, #A31570, Molecular Probes) secondary antibodies.

## Evaluation of RNA interacting with PQBP5 in AFM

RNAs contamination of protein samples was evaluated at multiple steps during purification of His-Tag PQBP5 protein for AFM. Briefly RNAs extracted from intermediate and final protein samples using RNeasy mini kits (74106, Qiagen) were loaded onto agar gels. To eliminate genomic DNA contamination, DNA was digested on-column digestion using DNase I during RNA isolation.

## RT-PCR

RNA was isolated from Rosetta™ 2(DE3) Singles™ Competent *E. coli* cells (#71400, Sigma Aldrich) or His-PQBP5 protein samples using RNeasy mini kits (74106, Qiagen). During this process, DNA was digested on-column with DNase I to eliminate genomic DNA contamination. Reverse transcription was performed using SuperScript VILO cDNA Synthesis kits (11754-250, Invitrogen), and PCR was performed on a T100 Thermal Cycler (Bio-Rad, Hercules, CA, U.S.A.) using TaKaRa Ex Taq (#RR001A, TaKaRa). The endogenous controls were idnT, hcaT, and cysG for mRNA, and leuT, serU, serW, and proK for tRNA. After mixed with loading buffer (#9157, TaKaRa), the samples were separated by 1% agarose gel electrophoresis in 1 × TAE buffer (0.04 M Tris−acetate, 1 mM EDTA).

The primer sequences were idnT, 5′-CTGTTTAGCGAAGAGGA-GATGC-3′ (forward) and 5′-ACAAACGGCGGCGATAGC-3′ (reverse); hcaT, 5′-GCTGCTCGGCTTTCTCATCC-3′ (forward) and 5′-CCAACCAC-GCAGACCAACC-3′ (reverse); cysG, 5′-TTGTCGGCGGTGGTGATGTC-3′ (forward) and 5′-ATGCGGTGAACTGTGGAATAAACG-3′ (reverse); leuT, 5′-GCGAAGGTGGCGGAATTGGTAGAC-3′ (forward) and 5′-TGGTGCG-AGGGGGGGGACTTGAAC-3′ (reverse); serU, 5′-GGAGAGATGCCGGAG-CGGCTGAAC-3′ (forward) and 5′-TGGCGGAGAGAGGGGGATTTGAAC-3′ (reverse); serW, 5′-GGTGAGGTGTCCGAGTGGCTGAAGT-3′ (forward) and 5′-GGCGGTGAGGGGGGGATTCGAAG-3′ (reverse); and proK, 5′-CGGTGATTGGCGCAGCCTGGTAGC-3′ (forward) and 5′-GAGGATTC-GAACCTCCGACCCCTT-3′ (reverse).

The amplification conditions consisted of an initial incubation at 55 °C for 10 min, followed by denaturation at 95 °C for 2 min, 45 cycles of denaturation at 95 °C for 10 s, and annealing and extension at 68 °C for 60 s.

## Confirmation of SNP/GWAS data and polyQ disease association

44,993 SNPs related to *PQBP5* were confirmed using NCBI SNP database (dbSNP: https://www.ncbi.nlm.nih.gov/snp/). To find their association with polyQ diseases, research papers including genome-wide association studies of polyQ diseases were searched using Pubmed (https://pubmed.ncbi.nlm.nih.gov/) with keywords "PolyQ and SNP", "Polyglutamine and SNP", "CAG repeat and SNP", "PolyQ and GWAS", "Polyglutamine and GWAS" or "CAG repeat and GWAS", and 140 research papers were collected. SNP or GWAS data attached to the papers were examined by custom code for relevance to 44,993 SNPs.

## Statistics

Statistical analyses for biological experiments were performed using Graphpad Prism 8 or Microsoft Excel for Microsoft 365.

Biological data following a normal distribution are presented as the mean ± SEM, with Tukey's HSD test utilized for multiple group comparisons.

The distribution of observed data was depicted with box plots, with the data also plotted as dots. Box plots show the medians, quartiles, and whiskers, which represent data outside the 25th–75th percentile range.

## Ethics

This study was performed in strict accordance with the recommendations of the Guide for the Care and Use of Laboratory Animals of the Japanese Government and the National Institutes of Health. All experiments were approved by The Committees for Gene Recombination Experiments and Animal Experiments of the Tokyo Medical and Dental University (G2018-082C3 and A2021-211A).

## Reporting summary

Further information on research design is available in the Nature Portfolio Reporting Summary linked to this article.

## Data availability

SNPs of PQBP5 were collected NCBI SNP database (https://www.ncbi.nlm.nih.gov/snp/). Search results of SNP/GWAS data and polyQ disease-associated research papers are available at https://doi.org/10.5281/zenodo.7389755. All other data generated or analyzed during this study are included in this article. Source data are provided with this paper.

## Code availability

A custom code for confirmation of SNP/GWAS data and polyQ disease association are available at https://doi.org/10.5281/zenodo.7389755[103].

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

## Acknowledgements

The authors thank Carl Zeiss Co., Ltd and Yokogawa Electric Corporation for their generous support regarding SRM. The authors also thank Dr. Xuemei Zhang (Tokyo Medical and Dental University) for mouse and histology experiments, Dr. Teikichi Ikura (Ochanomizu University) for advice on protein purification, and Drs. Noriki Katayama and Kazuma Tatsumi (Kanazawa University) for AFM. This work was supported by grants to H.O., including a Grant-in-Aid for Scientific Research on Innovative Areas (Foundation of Synapse and Neurocircuit Pathology, 22110001/ 22110002) from the Ministry of Education, Culture, Sports, Science, and Technology of Japan (MEXT), and a Grant-in-Aid for Scientific Research A (16H02655, 19H01042, 22H00464) from the Japanese Society for the Promotion of Science (JSPS).

## Author contributions

X.J., H.T., M.J., K.F., K.U., and N.K.: acquisition and analysis of data, and drafting of the manuscript. M.I. and H.Y.: acquisition of data, H.H. and T.A.: analysis of data and drafting of the manuscript. H.O.: conception and design of the study, obtaining funding, and drafting the manuscript.

## Competing interests

The authors declare no competing interests.
