## [Peer Review File · Nature Communications]

PQBP5/NOL10 maintains and anchors the nucleolus under physiological and osmotic stress conditionsREVIEWER COMMENTS

Reviewer #1 (Remarks to the Author):

In this manuscript, Jin, et al. utilize a combination of microscopic techniques (SRM, CLEM, and ATM) to demonstrate that PQBP5 (NOL10) contains an intrinsically disordered C-terminus and participates in the nucleolus by forming a skeletal meshwork in the nucleolar granular component (GC). Importantly, their studies establish a hierarchy whereby PQBP5 forms a framework for nucleolin, and fibrillarin, well-characterized proteins localized to the nucleolar GC and dense fibrillar component (DFC), respectively. Finally, using both biophysical assays and osmotic stress in cells, they showed that PQBP5 functions as an anchor for the reassembly of the nucleolus following stress. These findings are well-controlled, technically convincing, and provide novel findings important to those studying nucleolar dynamics. As such, I would support their publication in Nature Communications.

On the other hand, the authors finish by providing somewhat less-convincing data that PQBP5 co-sequestration by the polyglutamine disease proteins, Atxn1 and Htt, leads to the disappearance of the nucleolus. Although findings in U2OS cells (Supplementary figure 11) are striking, interaction data and in vivo studies are less convincing. Specifically,

1. There is high background in the IHC images in figure 5a making it difficult to distinguish specific signals. It would also help to show IHC of Atxn1 and HTT (red only) so that “sequestering” is less ambiguous. This data is less convincing than that in U2OS cells. In addition to making these data more convincing, I suggest moving this figure to the supplement and replacing it with more robust data presented in supplementary figure 11.

2. IP data in supplementary figure 14 extremely weak (especially between PQBP5 and Htt-17Q and Htt-103Q), not well controlled (no IPs of PQBP5 proteins in the absence of Atxn1 and Htt), and not completely consistent with SPR in supplementary figure 13.

Minor concerns:

Figure 4a - it is not immediately clear which graphs correlate with + vs – images. Could separate down the middle more.

Figure 5a - It is difficult to read the image labels, particularly the ones in red (Atxn1, Htt).

Discussion is cursory. It could describe a model of the findings and speculate about implications and suggestive data. For example, how the nucleolus plays a role in polyQ disease. Also, fibrillarlin appears to completely disappear from the cell in the presence of polyQ proteins but not, for example, with osmotic shock. Why?

Reviewer #2 (Remarks to the Author):

In this work, Jin et al., described the role of PQBP5 in the formation of the nucleolus. By using biochemical, structure, and cell biology methods, they showed that PQBP5 has an IDR domain that serves for the localization in nucleoli and that PQBP5-KD affects nucleoli structure. While the role of PQBP5 in nucleoli structure is interesting, the present manuscript failed to provide details for the performed analyses and does not discuss the data. Many results are presented just as a list of experiments without any clarification/discussion/conclusion. This is also evident by the half-page discussion at the end of the manuscript. Considering that the data were not all discussed in the Result section, the lack of a proper discussion makes this work very unclear. Further, the Introduction missed important points such as the description of PQBP5 and the relevant literature. On the other side, the Introduction contained a paragraph about nucleolus function in longevity that was not a topic of this work. In many parts, the authors are not precise and out of place. For example, the statement of the authors that “the knowledge about morphological and molecular changes under various cellular stress conditions remains extremely limited” is not entirely true. There are many studies, and none of them was cited. Further, why do they describe “only one study to date has addressed protein dynamics during heat shock and recovery, finding that misfolded nuclear proteins in nucleoplasm reversibly entered the nucleolus, especially GC, under heat shock conditions”? This manuscript did not deal with protein dynamics and heat shock.

Here some (not all) major criticism

The authors should better introduce PQBP5 and explained why they decided to study this factor.

Figure 1

Fig. 1b. Many important details are lacking. There is no information in the result section how AFM of PQBP5 was performed. At least they should mention that this is a recombinant protein. What is the RNA molecule in contact with PQBP5? What is the model head-to-tail interaction placed below the AFM of Fig. 1b?

Suppl. Figs. 1a,b. The interaction of PQBP5 with FBL and NCL are described with minimal terms and the conclusions are either too strong (PQBP5 might indirectly interact with fibrillarlin via nucleolin) or not

clear at all (Immunoprecipitation detected only a faint band containing any two of the three nucleolar proteins). Further, these data should be presented after Figs. 1d-h. It is very difficult to see the FC/DFC in the EM images and how the fibrillarin staining (SRM) matches. If the authors want to show this, they should modify Suppl. Fig. 4 and make these data visible.

At the end of this result section, the authors stated that the distribution pattern between NPM1 and PQBP5 differed completely. However, no data on NPM1 are shown.

Suppl. Figure 2

The authors should clarify what is NUC153 domain.

Fig. 2

At the beginning of this result section the authors should describe which proteins are targeted for shRNA-KD.

In the legend of Fig. 2c, the authors have to state the number of analyzed cells and the number of experiments (This is for all the data and corresponding quantifications)).

Lanes 218-219 "...complete depletion of PQBP5 resulted in the absence of all nucleoli". Technically, this cannot be stated since PQBP5 levels were not measured. The authors should indicate how they define low and high shRNA expression.

Lanes 225-226 "The deformities of the nucleoli were classified into three elements...". These alterations shown in Suppl. Fig. 7c should be described in the text. Further, the Venn diagrams shown in Suppl. Fig. 7c are not at all clear.

Lanes 216 "PQBP5 is the master regulator of nucleolus formation" This sentence should be tone down. Simply, PQBP5 is required for nucleolus formation

Lanes 231-233 ""PQBP5 is located at the top of the hierarchy that also includes nucleolin and fibrillarin". This statement should be corrected since the author analyzed only three proteins and, consequentially, this hierarchy cannot also include nucleolin and fibrillarin. The author should simply state that PQBP5 is essential for nucleolus formation.

Figure 3. The reason to analyze the function of PQBP5 under osmotic stress is not clear. Further, the link of osmotic stress to elderly people and during hyperglycemic has no reference. The authors describe only the data, without any conclusion. Several works have studied the nucleolus and its component under osmotic stress. The authors should cite these works and state whether their results are consistent or not consistent with the literature. They should also clarify why PQBP5 nucleolar localization is not affected.

Figure 4 shows that PQBP5 can form bigger droplets than fibrillarin or nucleoli. However, these analyses missed many important details such as salt concentration, the eventual presence of PEG, and the concentration of rRNA. Further, what rRNA has been used? In vitro synthesized rRNA? The whole 45S rRNA? Finally, Fig. 4c showed the structure of droplets containing PQBP5/Fibrillarin/Nucleoli. There is no comment or description of the Fibrillarin forming a ring around Nucleolin and PQBP5 (Fig. 4c). This is quite curious since in the cell nucleolus Fibrillarin is inside.

Suddenly, the authors investigated the role of PQBP5 in U2OS cells expressing Atxn1 mutant, which cause spinocerebellar ataxia type 1, and Huntingtin, which causes Huntington's disease. I decided to not go into the details of this analysis since I could not grab why this analysis was done and the conclusion of it.

Reviewer #3 (Remarks to the Author):

The authors have studied the nucleolar protein PQBP5 and its role in the morphology of nucleoli and how it interacts with and coordinates other nucleolar proteins. In addition, the authors studied the response and role of PQBP5 upon osmotic stress and upon expression of polyQ proteins.

The authors performed a tremendous amount of work and the imaging analyses of the mesh-work formed by PQBP5 and fibrillarin and nucleolin are impressive. They could further show the droplet formation of PQBP5 and have analysed these proteins by in silico techniques biochemical analyses of the interaction of the purified protein (SPR analyses). The data are sound, the imaging data of very high quality. They indeed advance our understanding of the sub-cellular localisation and interaction with other components of the nucleolus.

The weak point of this study is the lack of coherence in the "story" of the manuscript. How do osmotic stress and polyQ proteins fit together as stressors for nucleoli? These appear like two separate studies. I miss a clear motivation for the osmotic stress experiment. The statement that dehydration occurs in old people and those affected by diabetes is for me disconnected from an analysis of the nucleolus. What is the actual research question and what was their hypothesis.

The motivation to study polyQ proteins is a bit more obvious as PQBP5 is a polyQ binding protein. Yet, how does PQBP5 contribute to polyQ diseases? Are there mutations in PQBP5 that affect disease onset and pathology? This is not clear and it is hard to interpret the data presented here how they relate to polyQ diseases.

They end the section of the polyQ proteins (lines 337-339) with a statement: "Interestingly, normal and mutant Atxn1 and Htt proteins were found to bind similarly to PQBP5. Therefore, subsequent processes, such as degradation of the complexes with mutant proteins would contribute to the reduction of PQBP5"

First, how do the authors interpret the data that PQBP5 interacts with non-pathogenic and pathogenic polyQ proteins alike? Second, do PQBP5 protein levels change upon polyQ expression?

I miss also a control on the knockdown efficiencies of PQBP5

Minor comments:

1. What is the physiological relevance to perform thermal shift assays up to 100C or even beyond as discussed in lines 298-303?

2. Some statements should be revised e.g. line 198: "PQBP5 is at the top of the hierarchy and essential for nucleolus formation" Such a statement is not scientific.

Reviewer #4 (Remarks to the Author):

The manuscript, "PQBP5/NOL10 maintains and anchors the nucleolus under physiological and osmotic stress conditions" by Jin et al., describes a series of well-designed experiments aimed at understanding the structural role of PQBP5 in the nucleolus. The SRM data throughout the manuscript is quite convincing with regard to the distribution of PQBP5 in the nucleolus. Coupled with the different manipulations (knock-downs, etc.). The data is quite convincing with regard to the phenomenological role of PQBP5 in the nucleolus. If there is a weakness to the study, it is that there is a lack of quantitation to support some of the claims. This is an issue inherent to some of the techniques used. In general, I would be recommend the paper for publication.

- The high speed AFM was quite interesting; however, I had a few questions. Were the RNA and PQBP5 premixed in solution before deposition of the sample onto mica? If so, I would be curious to know the distribution of complexes. That is, how many free protein vs how many complexed with the RNA. The transient structure of the terminal end of PQBP5 interacting with the RNA in movie 3 is a little less convincing. Similar transient structure forming is observed in movie 1 without the RNA. It may just be that with random walk statistics that the terminal end just happens to touch the RNA in close proximity. Can some sort of statistical measure demonstrate that the end sticks to the RNA? I don't think this detracts from the main finding of the AFM study. Namely, that there is a flexible disordered chain.
- I am curious about the statement on line 337 with regard to PQBP5 bound similarly to Atxn1 and htt in both their normal and expanded forms. Do these data provide any information with regard to stoichiometry? I ask because some proteins (for example the MW1 antibody) that bind polyQ regions do so in a linear lattice type fashion. That is, if the polyQ region is expanded, more proteins can bind to it. I am curious if PQBP5 behaves in a similar way. This may enhance the sequestration phenomenon observed elsewhere in the paper with cells and mice.
- This may be beyond the scope of this initial study, but I am also curious about how PQBP5 interacts with different aggregation states of Atxn1 and htt. Does it have to interact with soluble species initially before being pulled into larger inclusions? I ask because ssNMR studies of htt fibrils indicate that the polyQ forms a dense core, and this often excludes polyQ binding proteins (again like antibodies). I think, based on the distribution of PQBP5 in the inclusions that it is being pulled into the inclusions during formation. It would be interesting to see if aggregation inhibitors that prevent inclusion formation would restore the proper distribution of PQBP5 and fibrillarin or if the longer polyQ stretch would sequester PQBP5 without inclusion formation. I think there is quite a bit of fertile ground here with implications for polyQ based diseases, and I look forward to seeing what future work will bring.
- Minor point: Supplemental Figure 13 is difficult to read with the thin lines present.

Reviewer's Comments:

Reviewer #1 (Remarks to the Author)

In this manuscript, Jin, et al. utilize a combination of microscopic techniques (SRM, CLEM, and ATM) to demonstrate that PQBP5 (NOL10) contains an intrinsically disordered C-terminus and participates in the nucleolus by forming a skeletal meshwork in the nucleolar granular component (GC). Importantly, their studies establish a hierarchy whereby PQBP5 forms a framework for nucleolin, and fibrillarin, well-characterized proteins localized to the nucleolar GC and dense fibrillar component (DFC), respectively. Finally, using both biophysical assays and osmotic stress in cells, they showed that PQBP5 functions as an anchor for the reassembly of the nucleolus following stress. These findings are well-controlled, technically convincing, and provide novel findings important to those studying nucleolar dynamics. As such, I would support their publication in Nature Communications.

>>> Thank you very much for kind evaluation.

On the other hand, the authors finish by providing somewhat less-convincing data that PQBP5 co-sequestration by the polyglutamine disease proteins, Atxn1 and Htt, leads to the disappearance of the nucleolus. Although findings in U2OS cells (Supplementary figure 11) are striking, interaction data and in vivo studies are less convincing. Specifically,

>>> We appreciate the criticism from the reviewer and improve the manuscript following the comments.

1. There is high background in the IHC images in figure 5a making it difficult to distinguish specific signals.

>>> We performed again IHC and now showed the images with lower background signals.

It would also help to show IHC of Atxn1 and HTT (red only) so that “sequestering” is less ambiguous. This data is less convincing that that in U2OS cells.

>>> We added IHC with red only (Atxn1 only or Htt only), which we hope support sequestration.

In addition to making these data more convincing, I suggest moving this figure to the supplement and replacing it with more robust data presented in supplementary figure 11.

>>> Following the comment, we made previous Supplementary 11 to Figure 6. Meanwhile, we improved previous Figure 5 following the advices of other reviewers. Considering the improved quality of the Figure 5, we are considering it might be better to keep it as Figure 7, and would like to ask the advice of the reviewer on this arrangement. Please advice whether we can keep it as a Figure or should move it Supplementary Figures.

2. IP data in supplementary figure 14 extremely weak (especially between PQBP5 and Htt-17Q and Htt-103Q), not well controlled (no IPs of PQBP5 proteins in the absence of Atxn1 and Htt), and not completely consistent with SPR in supplementary figure 13.

>>> In our experiences handling intrinsically disordered proteins, even though the two molecule form co-liquid droplets, it is difficult to show strong bands in immunoprecipitation. We do not know the mechanisms well but presumably, immunoprecipitation (IP) uses extract solutions with detergents like SDS or NP40, and they might destroy the LLPS-based droplets where the two proteins for IP analysis interact with each other.

In new Supplementary Figure 13b, we performed IP of PQBP5 proteins in the absence of Atxn1 and Htt, and IP of Atxn1 and Htt in the absence of PQBP5.

In contrast, SPR is more efficient to detect the interaction at a high affinity presumably because the buffer did not include SDS and because one of the two proteins is fixed on the surface of slide, mimicking Liquid-Liquid interaction surface.

Minor concerns:

Figure 4a - it is not immediately clear which graphs correlate with + vs – images. Could separate down the middle more.

>>> We added indications of RNA + or – above the graphs.

Figure 5a - It is difficult to read the image labels, particularly the ones in red (Atxn1, Htt).

>>> We modified the red color of Atxn1 and Htt and also make them bigger in their font size (previous Figure 5 becomes new Figure 6).

Discussion is cursory. It could describe a model of the findings and speculate about implications and suggestive data. For example, how the nucleolus plays a role in polyQ disease.

>>> We really appreciate the kind advices from the reviewer. In original submission, the paper has to be very short due to the limitation of words, while in current Nature Commun we can extend it. We followed the reviewer's advice and described about the points raised by the reviewer.

Also, fibrillarin appears to completely disappear from the cell in the presence of polyQ proteins but not, for example, with osmotic shock. Why?

>>> Thank you very much for thoughtful comments. This complete disappearance was due to the automatic modification of signals by the genuine soft ware integrated in microscopy. We adjusted the signals of fibrillarin in non-transfected cells in normal polyQ and mutant polyQ expressions. In the new image, fibrillarin is dispersed but not disappeared. However, it is also true that the signals of fibrillarin are weaker in polyQ expression than that in osmotic stress.

There are some differences in experimental situations between osmotic change stress and polyQ expression. According to a recent report, as the reviewer may know, hyperosmotic stress induced nuclear foci including proteasomal proteins and VCP (Yasuda et al, Nature 2020).

Yasuda, S., Tsuchiya, H., Kaiho, A. *et al.* Stress- and ubiquitylation-dependent phase separation of the proteasome. *Nature* **578**, 296–300 (2020).

They suggested that the LLPS foci function for protein degradation, based on co-relationship of proteasome inhibitor treatment and foci number/size or of ubiquitin inhibitor pre-treatment and foci formation. However, no direct proof of protein degradation is executed at LLPS foci. Conversely, it is possible that LLPS foci are a warehouse for proteasome proteins where they do not function. If it is the case, some proteins are not digested by proteasome under osmotic stress while they are degraded under mutant protein expression. Another possibility is a difference in time durations of experiments. Osmotic stress is observed for 24 hours and osmotic shock is only 20 min when proteasome system could be activated, while the effect of polyQ protein expression is observed after 72 hours from transfection. Even if the proteasome activation occurs at a similar level, proteins will be degraded more in the experiment of polyQ expression.

We discussed about this issue in Discussion.

Reviewer #2 (Remarks to the Author)

In this work, Jin et al., described the role of PQBP5 in the formation of the nucleolus. By using biochemical, structure, and cell biology methods, they showed that PQBP5 has an IDR domain that serves for the localization in nucleoli and that PQBP5-KD affects nucleoli structure. While the role of PQBP5 in nucleoli structure is interesting, the present manuscript failed to provide details for the performed analyses and does not discuss the data. Many results are presented just as a list of experiments without any clarification/discussion/conclusion.

This is also evident by the half-page discussion at the end of the manuscript. Considering that the data were not all discussed in the Result section, the lack of a proper discussion makes this work very unclear. Further, the Introduction missed important points such as the description of PQBP5 and the relevant literature. On the other side, the Introduction contained a paragraph about nucleolus function in longevity that was not a topic of this work.

>>> We appreciate critical comments on the paper writing. Following the suggestion we substantially increased the information about the details of analyses and discussion of the results in the manuscript. In addition, following the reviewer's advice for Introduction, we extensively rewrite Introduction of the manuscript. Regarding the paragraph about ribosome-longevity in Introduction, though we think it is indirectly related to the topic, we followed the comment and deleted the paragraph.

In many parts, the authors are not precise and out of place. For example, the statement of the authors that "the knowledge about morphological and molecular changes under various cellular stress conditions remains extremely limited" is not entirely true. There are many studies, and none of them was cited.

>>> We increased the number of references from 37 papers to 93 papers, and focused on the background information about "the knowledge about morphological and molecular changes under various cellular stress conditions" in the field. If the reviewer considers some specific paper(s) related to the topic, we would be glad to refer them if reasonable.

Further, why do they describe “only one study to date has addressed protein dynamics during heat shock and recovery, finding that misfolded nuclear proteins in nucleoplasm reversibly entered the nucleolus, especially GC, under heat shock conditions”? This manuscript did not deal with protein dynamics and heat shock.

>>> We deleted this part.

Here some (not all) major criticism

The authors should better introduce PQBP5 and explained why they decided to study this factor.

>>> We described about the reason in Introduction.

>>> In “Introduction”, we described more about our research history and strategy. Along with the history, we described that we have been searching for key molecules by various comprehensive analyses, that most of the identified key molecules are IDPs, that we are analyzing the functions of key molecules in neurodegeneration, that we are screening the effects of various stresses on key IDPs, and that we noticed that osmotic stress affects PQBP5 which binds to polyQ proteins.

Figure 1

Fig. 1b. Many important details are lacking. There is no information in the result section how AFM of PQBP5 was

performed. At least they should mention that this is a recombinant protein.

>>> We agree that previous Method section of “protein purification” was confusing. We divided the section to two new sections. One is “Sample preparation for high-speed AFM and Droplet assay” and another is “Protein preparation for SPR analysis”. We put these sections in front of the section for “AFM” and “SPR analysis”.

As you can see in Methods of the previous version of manuscript, we used His-Tagged PQBP5 for AFM.

What is the RNA molecule in contact with PQBP5?

>>> We examined co-purified RNAs (PQBP5-interacting RNAs) by agar gel (Supplementary Figure 14). The band of 23S or 16S ribosomal RNA was hardly detected while low molecular smears around 1.0kb and below 0.5kb were faintly observed, indicating that the RNA molecule would be small bacterial rRNAs such as 5S (counterpart of 5.8S mammalian rRNA). Considering with the molecular size of RNA relative to 688 amino acid of PQBP5, 5S bacterial rRNA is one of the best candidates. However, further study defining the details of PQBP5-interacting RNAs would be a next project. We discussed about the speculation together with the results of LLPS with rRNA in Discussion.

What is the model head-to-tail interaction placed below the AFM of Fig. 1b?

>>> We explained the models in Figure 1b legend.

Suppl. Figs. 1a,b. The interaction of PQBP5 with FBL and NCL are described with minimal terms and the conclusions are either too strong (PQBP5 might indirectly interact with fibrillarin via nucleolin) or not clear at all (Immunoprecipitation detected only a faint band containing any two of the three nucleolar proteins).

>>> We changed the expressions on these results of IP, and described together with the results of SPR.

Further, these data should be presented after Figs. 1d-h.

>>> We moved Suppl. Figs. 1a,b. after Figs. 1d-h.

It is very difficult to see the FC/DFC in the EM images and how the fibrillarin staining (SRM) matches. If the authors want to show this, they should modify Suppl. Fig. 4 and make these data visible.

>>> We added three new nuclei having obvious FC/DFC, and showed fibrillarin stains matched well with DFC (Supplementary Figure 3). Here, we again showed the same nucleus of Figure 1h (shown in an merged color) in separated colors, and revealed that fibrillarin stains matched well with DFC also in this nucleolus (Supplementary Figure 3).

At the end of this result section, the authors stated that the distribution pattern between NPM1 and PQBP5 differed completely. However, no data on NPM1 are shown.

>>> We added new results of super resolution microscopy showing the relationship between PQBP5, NPM1 and PES1 (Supplementary Figure 5).

Suppl.Figure 2

The authors should clarify what is NUC153 domain.

>>> We described its information in Result section of the text.

Fig. 2

At the beginning of this result section the authors should describe which proteins are targeted for shRNA-KD.

>>> We followed the advice and wrote the targets in the first sentence of this section.

In the legend of Fig.2c, the authors have to state the number of analyzed cells and the number of experiments (This is for all the data and corresponding quantifications)).

>>> We showed the number of analyzed cells and the number of experiments in all figures.

Lanes 218-219 "...complete depletion of PQBP5 resulted in the absence of all nucleoli". Technically, this cannot be stated since PQBP5 levels were not measured. The authors should indicate how they define low and high shRNA expression.

>>> We followed the advice and deleted the phrase "...complete depletion of PQBP5 resulted in the absence of all

nucleoli”. We also defined the low and high shRNA expression levels by the density of gold particles.

Lanes 225-226 “The deformities of the nucleoli were classified into three elements...”. These alterations shown in Suppl. Fig. 7c should be described in the text. Further, the Venn diagrams shown in Suppl. Fig. 7c are not at all clear.

>>> In previous version of manuscript, we defined abnormal nucleoli with three elements, irregular shape, no DFC, and low density. Then we defined three categories of abnormal nucleoli by Venn diagram, abnormal nucleoli having more than one element, abnormal nucleoli having more than two elements, and abnormal nucleoli having three elements.

However, these definitions and analyses based on the categorization might be too complicated. Therefore, in the new version of manuscript, we showed examples of abnormal nucleoli and showed the percentage of abnormal cells having abnormal nucleoli with two or more abnormal features.

Lanes 216 “PQBP5 is the master regulator of nucleolus formation” This sentence should be tone down. Simply, PQBP5 is required for nucleolus formation

>>> We followed the advice, toned down the sentence, and described “PQBP5 has the largest effect on nucleolus formation among three proteins”.

Lanes 231-233 ““PQBP5 is located at the top of the hierarchy that also includes nucleolin and fibrillarin”. This statement should be corrected since the author analyzed only three

proteins and, consequentially, this hierarchy cannot also include nucleolin and fibrillarin. The author should simply state that PQBP5 is essential for nucleolus formation.

>>> We agree with the criticism that we analyzed three proteins and they are not all the nucleolar proteins.

Meanwhile, the evaluation from reviewer #1 is contradictory to the evaluation of reviewer #1 in which the hierarchy is highly appreciated. This high evaluation is also consistent with our idea coming from both functional analysis and structural analysis suggesting PQBP5's nest-like structure in which DFC/FCs are embraced. NPM1 and PES1, which were analyzed in the revised manuscript, are wrapping the nest structure of PQBP5. All the data suggested the dominance of PQBP5 in the function of nucleolus formation.

Therefore, we changed the expression weaker and wrote "... suggest that PQBP5 is the most dominant regulator among three IDPs (PQBP5, fibrillarin and nucleolin)".

Figure 3. The reason to analyze the function of PQBP5 under osmotic stress is not clear. Further, the link of osmotic stress to elderly people and during hyperglycemic has no reference. The authors describe only the data, without any conclusion.

>>> Dehydration in elderly people is the common sense for clinical doctors, so we did not mention the references and we apologize for unkindness. Following the advice, we attached here some references. Also hyperglycemia and dehydration is

the common sense for clinical doctors, but we attached the reference following the advice.

>>> In “Introduction”, we described that we have been searching for key molecules by various comprehensive analyses, that most of the identified key molecules are IDPs, that we are analyzing the functions of key molecules in neurodegeneration, that we are screening the effects of various stresses on key IDPs, and that we noticed that osmotic stress affects PQBP5 that has been known to bind to polyQ proteins from previous studies.

From the viewpoint of protein science, it is natural to test characteristics of proteins under various environmental factors. This is a fundamental reason. Another reason is PQBP5 interacts with polyQ proteins, and in order to know the basis of pathology derived from polyQ-PQBP5 pathological interaction, we need to know firstly what is the physiological function of PQBP5. Beginning from the fundamental question for protein science, we first know PQBP5 is an anchor protein of nucleolus (mainly from osmotic experiments and knockdown experiments), next we know mutant polyQ induce the PQBP5 deficiency from nucleolus by “sequestration” to polyQ inclusion body, and then we know mutant polyQ causes nucleolar loss/abnormality via deficiency of PQBP5 in nucleolus. This could be a conclusion, we believe.

Several works have studied the nucleolus and its component under osmotic stress. The authors should cite these works and state whether their results are consistent or not consistent with the literature.

>>> We increased the number of references from 37 papers to 93 papers, and mostly focused on the background information about “the knowledge about morphological and molecular changes under various cellular stress conditions” in the field. As far as we know, we could not reach to papers that exactly match to the comment of this reviewer. If the reviewer considers some specific paper(s) related to the topic, we would be glad to refer them if reasonable. So please indicate them.

They should also clarify why PQBP5 nucleolar localization is not affected.

>>> That is the reason why we performed thermal shift and thermal x osmotic shift assays in new Figure 5 (previous Supplementary Figure 10). PQBP5 is structurally most stable among three proteins. Since this figure is important in the story as the reviewer questioned, we moved it to new Figure 5.

Figure 4 shows that PQBP5 can form bigger droplets than fibrillarin or nucleoli. However, these analyses missed many important details such as salt concentration, the eventual presence of PEG, and the concentration of rRNA.

>>> In previous version of manuscript, we had already described the information (salt concentration, the eventual presence of PEG, and the concentration of rRNA) in Methods. We did not use Crowders such as PEG in our droplet experiments. Sorbitol for high osmolarities might have affected the droplet formation, while it was not the case (no remarkable difference in droplet formation).

Further, what rRNA has been used? In vitro synthesized rRNA?
The whole 45S rRNA?

>>> For droplet assay, we prepared rRNA from HeLa cells and all rRNA including 28S, 18S and other small rRNAs.

Finally, Fig. 4c showed the structure of droplets containing PQBP5/Fibrillarin/Nucleoli. There is no comment or description of the Fibrillarin forming a ring around Nucleolin and PQBP5 (Fig. 4c). This is quite curious since in the cell nucleolus Fibrillarin is inside.

>>> We appreciate the comment. We have noticed this finding and thought it very interesting. This is the physical fact and definitely true, so we need a hypothesis and we had hesitated to write such a speculation. But we followed the comment from the reviewer and wrote the speculation in revised version.

Suddenly, the authors investigated the role of PQBP5 in U2OS cells expressing Atxn1 mutant, which cause spinocerebellar ataxia type 1, and Huntingtin, which causes Huntington's disease. I decided to not go into the details of this analysis since I could not grab why this analysis was done and the conclusion of it.

>>> We described in "Introduction" why these analyses were done in this study. The reasons were already described in responses to the comment on Figure 3.

The former part of this paper is not only for osmotic stress but also for essential function of PQBP5 for nucleolar structure that is indispensable for understanding pathological functions of

polyQ proteins mediated by PQBP5. Consistently with the popular hypothesis supported by a number of publications in the research field of polyQ diseases, sequestration of PQBP5 to polyQ aggregates, which mimics the knockdown experiments, resulted in the loss or abnormal structures of nucleolus.

We have considered about deleting the latter part (pathological side) of this manuscript and making the manuscript only for the physiological side of PQBP5, as the reviewer remark the strong comment for the latter part. However, with positive evaluations and encouragements on the latter part from other reviewers, we think it would be more valuable for readers and science to keep the latter part and to present the former (physiological side) and the latter (pathological side) parts in combination. We would like to follow the editor's advice about the issue of the paper structure.

Reviewer #3 (Remarks to the Author)

The authors have studied the nucleolar protein PQBP5 and its role in the morphology of nucleoli and how it interacts with and coordinates other nucleolar proteins. In addition, the authors studied the response and role of PQBP5 upon osmotic stress and upon expression of polyQ proteins.

The authors performed a tremendous amount of work and the imaging analyses of the mesh-work formed by PQBP5 and fibrilarin and nucleolin are impressive. They could further show the droplet formation of PQBP5 and have analysed this proteins by in silico techniques biochemical analyses of the interaction of

the purified protein (SPR analyses). The data are sound, the imaging data of very high quality. They indeed advance our understanding of the sub-cellular localisation and interaction with other components of the nucleolus.

>>> Thank you very much for kind evaluation from the reviewer.

The weak point of this study is the lack of coherence in the "story" of the manuscript. How do osmotic stress and polyQ proteins fit together as stressors for nucleoli? These appear like two separate studies. I miss a clear motivation for the osmotic stress experiment. The statement that dehydration occurs in old people and those affected by diabetes is for me disconnected from an analysis of the nucleolus. What is the actual research question and what was their hypothesis.

>>> We described in Introduction that we have been searching for key molecules in neurodegeneration and consequently found many IDPs as key molecules and that we are screening the effects of various stresses on key IDPs in neurodegeneration selected from our comprehensive big data approaches, and that we noticed that osmotic stress affects PQBP5 which binds to polyQ proteins and thereby influence polyQ pathologies.

We appreciate the criticism that the link from the first part to the second part is not well written. The relationship of the two parts is not "parallel" but "serial". Though the first part describes PQBP5-dependent resilience of nucleolus to osmotic stress, the main claim in the first part is that PQBP5 maintains nucleolus both in the physiological condition (for instance, KD experiment) and pathological condition (Osmotic stress). On this basis of the

first part, we examined, as the second step, the effect of PQBP5-deprivation by polyQ proteins. We explained this link in the first section of the second part in Result.

The motivation to study polyQ proteins is a bit more obvious as PQBP5 is a polyQ binding protein. Yet, how does PQBP5 contribute to polyQ diseases? Are there mutations in PQBP5 that affect disease onset and pathology? This is not clear and it is hard to interpret the data presented here how they relate to polyQ diseases.

>>> Again, I would repeat the same explanation. The “sequestration hypothesis” is one of the most popular and still surviving theories in the field of polyQ diseases, and it could be also the case in some other neurodegenerative diseases. Sequestration caused deprivation of PQBP5 from nucleolus, which is equivalent to KO or KD of PQBP5 in regards of the nucleolar function. Therefore, it is easy to speculate, if PQBP5 interacts with polyQ disease proteins, the sequestration of PQBP5 to immobile or aggregated foci (or inclusion body) will reduce the amount of PQBP5 in its functional region in cell (nucleolus), and it will lose the fundamental function of PQBP5 to maintain and anchor nucleolus. We explained this line of mechanistic insight in the first section of the second part (polyQ part) in Result.

As far as we know there is one paper suggesting relationship between PQBP5 and disease, and we referred the paper in Introduction. The relationship between human degenerative diseases and PQBP5 has not been reported so far. It would be an open question.

Such thing will occur, we believe. When we published impairment of DNA damage repair in polyQ pathology based on basic science experiments (Qi et al, Nat Cell Biol 2007; Enokido et al, JCB 2010) there had been no clinical data, while after 10 years it is now common knowledge that SNPs or mutations of DNA damage repair genes affect disease onset and pathology, as the reviewer may know well.

They end the section of the polyQ proteins (lines 337-339) with a statement: "Interestingly, normal and mutant Atxn1 and Htt proteins were found to bind similarly to PQBP5. Therefore, subsequent processes, such as degradation of the complexes with mutant proteins would contribute to the reduction of PQBP5"

First, how do the authors interpret the data that PQBP5 interacts with non-pathogenic and pathogenic polyQ proteins alike? Second, do PQBP5 protein levels change upon polyQ expression?

>>> We did not stress on the main explanation, "sequestration" here. Even if PQBP5 binds to normal polyQ proteins, it is not sequestered because no insoluble or less mobile inclusion body exists with normal polyQ proteins. After the first explanation, we describe the second explanation by protein degradation after interaction of mutant polyQ proteins and PQBP5.

I miss also a control on the knockdown efficiencies of PQBP5

>>> In Supplementary Figure 8a, we estimated KD efficiencies of PQBP5, by sh-PQBP5, sh-scramble and sh-RNA for other two proteins, in western blot.

Minor comments:

1. What is the physiological relevance to perform thermal shift assays up to 100C or even beyond as discussed in lines 298-303?

>>> We do not know why the authors of the reference performed thermal shift assay in such a high temperature that does not occur in living cells or animals. However, the information is useful from the aspect of chemists, and I also think it is important for understanding proteins physical chemistry.

2. Some statements should be revised e.g. line 198: "PQBP5 is at the top of the hierarchy and essential for nucleolus formation" Such a statement is not scientific.

>>> We accept that we cannot scientifically conclude it from our data handling only a limited number of proteins. In the new version, we wrote that PQBP5 is the most dominant regulator among three IDPs (PQBP5, fibrillarin and nucleolin). We hope that this description is acceptable for the reviewer.

Reviewer #4 (Remarks to the Author):

The manuscript, "PQBP5/NOL10 maintains and anchors the nucleolus under physiological and osmotic stress conditions" by Jin et al., describes a series of well-designed experiments aimed at understanding the structural role of PQBP5 in the nucleolus. The SRM data throughout the manuscript is quite convincing with regard to the distribution of PQBP5 in the nucleolus. Coupled with the different manipulations (knock-downs, etc.). The data is quite convincing with regard to the phenomenological role of PQBP5 in the nucleolus. If there is a weakness to the study, it is that there is a lack of quantitation to support some of the claims. This is an issue inherent to some of the techniques used. In general, I would be recommend the paper for publication.

- The high speed AFM was quite interesting; however, I had a few questions. Were the RNA and PQBP5 premixed in solution before deposition of the sample onto mica? If so, I would be curious to know the distribution of complexes. That is, how many free protein vs how many complexed with the RNA.

>>> In AFM of Figure 1, PQBP5's interaction with RNA, which was unexpectedly contaminated in the protein sample, was accidentally discovered. In this sense, RNA and PQBP5 were premixed before deposition of sample onto mica.

We evaluated the species and the amount of contaminated RNA. We identified the contaminated RNA to be degraded ribosomal RNA, and that the concentration of rRNA was 2.5 $\mu\text{g} / \text{mL}$ in the protein sample. We also know that the concentration of PQBP5's was 500 nmol /L. Together with the mean MW of degraded rRNA, the concentration of rRNA was calculated to be 9.7 nmol / L, and the mol ratio between PQBP5 and rRNA was roughly 50 : 1. In this

condition, we observed fourteen PQBP5 molecules, we found two molecules were bound and one molecule was un-bound with rRNA. We described these results in Discussion of the text.

We recognize that further studies are necessary for dynamics between PQBP5 and rRNA, but it will be done in the future.

The transient structure of the terminal end of PQBP5 interacting with the RNA in movie 3 is a little less convincing. Similar transient structure forming is observed in movie 1 without the RNA. It may just be that with random walk statistics that the terminal end just happens to touch the RNA in close proximity. Can some sort of statistical measure demonstrate that the end sticks to the RNA? I don't think this detracts from the main finding of the AFM study. Namely, that there is a flexible disordered chain.

>>> We appreciate the reviewer's thoughtful comment. We agree with the opinion of the reviewer that transient formation of small structure in IDP region is independent of RNA. The structure is generated stochastically, and it binds to RNA if RNA exists in neighborhood at the timing. Of course, if RNA does not exist in neighborhood, the structure cannot bind to RNA.

In response to the request of the reviewer for statistical measurement, we counted every one sec of movies during which whether PQBP5 forms the transient structure and whether PQBP5 contacts with RNA in two images ($N=2$, $n=35$), and thereby examined whether the transient structure is related to interaction. In both cells, Fischer's exact test suggested that the transient structure is related to the PQBP5-RNA binding in both cells.

- I am curious about the statement on line 337 with regard to PQBP5 bound similarly to Atxn1 and htt in both their normal and expanded forms. Do these data provide any information with regard to stoichiometry? I ask because some proteins (for example the MW1 antibody) that bind polyQ regions do so in a linear lattice type fashion. That is, if the polyQ region is expanded, more proteins can bind to it. I am curious if PQBP5 behaves in a similar way. This may enhance the sequestration phenomenon observed elsewhere in the paper with cells and mice.

>>> Previous Supplementary Figure 14 was revised to New Supplementary Figure 13 by adding new IP data (Supplementary Figure 13b). Previous IP were done with different filters for normal and expanded polyQ proteins, so they were not strictly comparable. The new data added to revision was performed in the same filter by which we can compare the affinity of PQBP5 to normal and expanded polyQ proteins. This new result revealed that expanded Atxn1 binds to PQBP5 stronger than normal Atxn1. In the case of Htt, normal Htt did not bind to PQBP5 in immunoprecipitation with transfected cells, because the affinity was low and because their localizations in subcellular domains were different. Collectively, the new results were consistent with the expectation of the reviewer.

Accordingly, we deleted the previous statement on line 337, and now described the fact more precisely, which is actually as the reviewer expected.

- This may be beyond the scope of this initial study, but I am also curious about how PQBP5 interacts with different aggregation states of Atxn1 and htt. Does it have to interact with soluble species initially before being pulled into larger inclusions? I ask because ssNMR studies of htt fibrils indicate that the polyQ forms a dense core, and

this often excludes polyQ binding proteins (again like antibodies). I think, based on the distribution of PQBP5 in the inclusions that it is being pulled into the inclusions during formation. It would be interesting to see if aggregation inhibitors that prevent inclusion formation would restore the proper distribution of PQBP5 and fibrillarin or if the longer polyQ stretch would sequester PQBP5 without inclusion formation. I think there is quite a bit of fertile ground here with implications for polyQ based diseases, and I look forward to seeing what future work will bring.

>>> We thank the reviewer again for this interesting and thoughtful comment. Hearing the speculation, we also consider that some soluble species of Atxn1 and Htt would interact with PQBP5 before they form dense core of fibrils. In this regard, aggregation inhibitors, dependently on their target state during aggregation, would affect interaction between PQBP5 and polyQ proteins. As suggested by the reviewer, this issue should be carefully investigated with abundant data, so we would keep the advised issue for a next research project in the near future. We borrowed this viewpoint of the reviewer and added it to Discussion of our manuscript.

- Minor point: Supplemental Figure 13 is difficult to read with the thin lines present.

>>> We changed the line thickness following the advice.

REVIEWER COMMENTS

Reviewer #1 (Remarks to the Author):

Consistent with my initial review, the findings in the revised manuscript are well-controlled, technically convincing, and provide novel findings important to those studying nucleolar dynamics. My support for their publication in Nature Communications was previously dependent upon addressing less-convincing data that PQBP5 co-sequestration by the polyglutamine disease proteins leads to the disappearance of the nucleolus. The revised manuscript has addressed all of my original concerns and I therefore support its publication as this will be of importance to researchers studying the nucleolus as well as those studying pathomechanisms in neurodegeneration.

- As there are some errors in sentence structure in the new expanded discussion, I feel that additional editing would make it better.

- In response to the author inquiry regarding the new figure 7, I agree that it now fits well as a main figure.

Reviewer #3 (Remarks to the Author):

Unfortunately, the revised version of the manuscript is now harder to read and understand as the original submission.

The authors expanded all sections by increasing the number of references and simply adding a lot of text. But they did not address the key issues I had. It is in this form even less focused and convincing.

1.:

my previous comment:

The weak point of this study is the lack of coherence in the "story" of the manuscript. How do osmotic stress and polyQ proteins fit together as stressors for nucleoli? These appear like two separate studies. I miss a clear motivation for the osmotic stress experiment. The statement that dehydration occurs in old people and those affected by diabetes is for me disconnected from an analysis of the nucleolus. What is the actual research question and what was their hypothesis.

answer by the authors:

>>> We described in Introduction that we have been searching for key molecules in neurodegeneration and consequently found many IDPs as key molecules and that we are screening the effects of various stresses on key IDPs in neurodegeneration selected from our comprehensive big data approaches, and that we noticed that osmotic stress affects PQBP5 which binds to polyQ proteins and thereby influence polyQ pathologies.

We appreciate the criticism that the link from the first part to the second part is not well written. The relationship of the two parts is not "parallel" but "serial". Though the first part describes PQBP5-dependent resilience of nucleolus to osmotic stress, the main claim in the first part is that PQBP5 maintains nucleolus both in the physiological condition (for instance, KD experiment) and pathological condition (Osmotic stress). On this basis of the first part, we examined, as the second step, the effect of PQBP5-deprivation by polyQ proteins. We explained this link in the first section of the second part in Result.

answer from me:

The revised version does not address my criticism. I am not sure the authors understand my point.

2.:

my previous comment:

The motivation to study polyQ proteins is a bit more obvious as PQBP5 is a polyQ binding protein. Yet, how does PQBP5 contribute to polyQ diseases? Are there mutations in PQBP5 that affect disease onset and pathology? This is not clear and it is hard to interpret the data presented here how they relate to polyQ diseases.

answer by the authors:

>>> Again, I would repeat the same explanation. The “sequestration hypothesis” is one of the most popular and still surviving theories in the field of polyQ diseases, and it could be also the case in some other neurodegenerative diseases. Sequestration caused deprivation of PQBP5 from nucleolus, which is equivalent to KO or KD of PQBP5 in regards of the nucleolar function. Therefore, it is easy to speculate, if PQBP5 interacts with polyQ disease proteins, the sequestration of PQBP5 to immobile or aggregated foci (or inclusion body) will reduce the amount of PQBP5 in its functional region in cell (nucleolus), and it will lose the fundamental function of PQBP5

to maintain and anchor nucleolus. We explained this line of mechanistic insight in the first section of the second part (polyQ part) in Result.

As far as we know there is one paper suggesting relationship between PQBP5 and disease, and we referred the paper in Introduction. The relationship between human degenerative diseases and PQBP5 has not been reported so far. It would be an open question.

Such thing will occur, we believe. When we published impairment of DNA damage repair in polyQ pathology based on basic science experiments (Qi et al, Nat Cell Biol 2007; Enokido et al, JCB 2010) there had been no clinical data, while after 10 years it is now common knowledge that SNPs or mutations of DNA damage repair genes affect disease onset and pathology, as the reviewer may know well.

answer from me:

The authors again failed to see my point. I specifically asked for known links between PQBP5 and polyQ such as mutations, GWAS studies etc. Why do they not address this question in the manuscript and quote here in the rebuttal letter where exactly they addressed my question. Because to me this has not been addressed and a response like "while after 10 years it is now common knowledge that SNPs or mutations of DNA damage repair genes affect disease onset and pathology, as the reviewer may know well."

is not acceptable.

3.:

my previous comment:

They end the section of the polyQ proteins (lines 337-339) with a

statement: "Interestingly, normal and mutant Atxn1 and Htt proteins were found to bind similarly to PQBP5. Therefore, subsequent processes, such as degradation of the complexes with mutant proteins would contribute to the reduction of PQBP5"

First, how do the authors interpret the data that PQBP5 interacts with non-pathogenic and pathogenic polyQ proteins alike?

Second, do PQBP5 protein levels change upon polyQ expression?

answer by the authors:

>>> We did not stress on the main explanation, "sequestration" here. Even if PQBP5 binds to normal polyQ proteins,

my response:

what do you mean by "even if PQBP5 binds..." you stated that PQBP5 binds to non-pathogenic and pathogenic polyQ proteins

Reviewer #4 (Remarks to the Author):

The authors responded to my criticisms of the initial submission well. In particular, I appreciate the statistical analysis of the high speed AFM. As a result, I would support publication based on my expertise.

Reviewer #5 (Remarks to the Author):

In this manuscript Jin et al describe the nucleolar localisation of PQBP5, identify the C terminus of the protein as being intrinsically disordered and suggest that the protein is key to maintaining the structure of the nucleolus. They also suggests it acts as an anchor for re-assembly of nucleolar structures.

The authors addressed some of the major concerns of reviewer 2 by adding more text to the introduction, results and discussion and citing more papers. However, their additional text required more focus and detail. Especially in relation to previous studies on nucleolar morphology under stress and the role of nucleolar stress in neurodegeneration.

They did clarify the specific points reviewer 2 had about the data and included all quantification details as suggested.

“...complete depletion of PQBP5 resulted in the absence of all nucleoli”. This phrase was removed but the statement that PQBP5 is the master regulator of nucleoli remained in the discussion.

Inhibition of PolI-driven transcription will cause relocalisation of nucleolar proteins into the nucleoplasm/cytoplasm and loss of nucleolar integrity. Therefore, it may be that the consequences of depleting PQBP5 are due to loss of 47S transcription, which could be independent from a structural role. Given that that PQBP5 binds RNA, the effects of PQBP5 loss on 47S transcription should be confirmed.

The DAPI channel was not shown in the images in figure 2 and so, it was difficult to see what happened to nuclei and nucleoli (using loss of DAPI staining as a marker for nucleoli rather than Fibrillarin and nucleolin which are altered in stress).

REVIEWER COMMENTS

※ We submit two types of manuscript file with or without edit history. In the version with edit history, we kept edit history of previous R1 revision with blue letters, showed newly edited parts in R2 revision with blue letters marked yellow, and also showed changes by professional English editor.

Reviewer #1 (Remarks to the Author):

Consistent with my initial review, the findings in the revised manuscript are well-controlled, technically convincing, and provide novel findings important to those studying nucleolar dynamics. My support for their publication in Nature Communications was previously dependent upon addressing less-convincing data that PQBP5 co-sequestration by the polyglutamine disease proteins leads to the disappearance of the nucleolus. The revised manuscript has addressed all of my original concerns and I therefore support its publication as this will be of importance to researchers studying the nucleolus as well as those studying pathomechanisms in neurodegeneration.

>>> Thank you very much for kind evaluation. We really appreciate the reviewer's kind support.

- As there are some errors in sentence structure in the new expanded discussion, I feel that additional editing would make it better.

>>> Following the advice, we received professional English editing of this second revision manuscript.

- In response to the author inquiry regarding the new figure 7, I agree that it now fits well as a main figure.

>>> Thank you very much.

Reviewer #3 (Remarks to the Author):

Unfortunately, the revised version of the manuscript is now harder to read and understand as the original submission. The authors expanded all sections by increasing the number of references and simply adding a lot of text.

>>> We appreciate very much the great efforts of reviewer #3. In this R2 revision, we tried to improve readability following the reviewer's comments, and asked again professional English editing of the R2 manuscript. We hope, this process improves the manuscript, and makes our idea clear in more concise and understandable descriptions. On the other hand, we appreciate kind understanding of the reviewer #3 that elongated parts were responses to other reviewers and those reviewers were basically satisfied with our R1 responses, and that we need to keep these parts in R2 revision.

But they did not address the key issues I had. It is in this form even less focused and convincing.

>>> Regarding the issue of disconnection between former part and latter part of the manuscript, the other reviewers raised the same point, while they are satisfied with our responses. To focus on our “research question” requested by reviewer #3, we improved one sentence in Introduction and added one paragraph in Discussion, as described in the following.

>>> Regarding the second issue of GWAS study, we did respond to the GWAS issue in previous R1 revision (line 119 – 123 of previous R1 manuscript). However, we did not mention about the description in rebuttal letter comment to reviewer #3, and also the color of letters was not change blue in this sentence. We apologize for these careless mistakes that might have confused reviewer #3 and prevented him/her to recognize our response in R1 revision.

In R2 revision, we kept the description and moreover added our search of PQBP5-associated SNPs with published databases in Introduction, as following.

1.:

my previous comment:

The weak point of this study is the lack of coherence in the "story" of the manuscript. How do osmotic stress and polyQ proteins fit together as stressors for nucleoli? These appear like two separate studies. I miss a clear motivation for the osmotic stress experiment. The statement that dehydration occurs in old people and those affected by diabetes is for me disconnected from an analysis of the nucleolus. What is the actual research question and what was their hypothesis.

answer by the authors:

>>> We described in Introduction that we have been searching for key molecules in neurodegeneration and consequently found many IDPs as key molecules and that we are screening the effects of various stresses on key IDPs in neurodegeneration selected from our comprehensive big data approaches, and that we noticed that osmotic stress affects PQBP5 which binds to polyQ proteins and thereby influence polyQ pathologies. We appreciate the criticism that the link from the first part to the second part is not well written. The relationship of the two parts is not “parallel” but “serial”. Though the first part describes PQBP5-dependent resilience of nucleolus to osmotic stress, the main claim in the first part is that PQBP5 maintains nucleolus both in the physiological condition (for instance, KD experiment) and pathological condition (Osmotic stress). On this basis of the first part, we examined, as the second step, the effect of PQBP5-deprivation by polyQ proteins. We explained this link in the first section of the second part in Result.

answer from me:

The revised version does not address my criticism. I am not sure the authors understand my point.

>>> “Our research question in this study” is to reveal the physiological and pathological functions of PQBP5 that was discovered as a binding protein to polyQ-tract sequence by yeast two-hybrid method (Imafuku et al, BBRC 1998). Originally in 1998, we hypothesize that proteins binding to polyQ sequence could be disease modifiers of polyQ diseases. But before revealing the pathological function of PQBP5, because PQBP5 is a new protein

whose physiological characteristics nobody knows, we needed to characterize the biophysics and functions of PQBP5.

In order to characterize the molecular features of such an intrinsically disordered protein that did not have a fixed 3D structure in the disordered region, it is necessary to know biophysical features of the molecule under osmotic stress, heat stress, and so on. We therefore checked them in Figure 3 and 5, respectively. Extent of change of each IDP under such stresses is important basic knowledge for what features the IDP has as a molecule, what kind of roles the IDP has in cells, and for how easily the IDP forms aggregates.

Nucleus is formed by multiple IDPs in addition to PQBP5. Their biophysics features would be different under various physical stresses, and such differences generate specific roles of each IDP proteins in nucleolus. Based on this research question, we perform multiple experiments and reveal that PQBP5 is relatively stable molecule in nucleus under osmotic stress, and also stable under heat stress.

The knowledge led to the next functional analysis, KD experiment. In the deficiency of PQBP5, nucleolus cannot keep the structure, indicating PQBP5 is one of the essential molecules for formation of nucleolus.

On the basis of elucidated biophysics and functions of PQBP5, going back to our original hypothesis at the time point of discovery of PQBP5 in 1998, we started analysis to know how this molecule could participate in polyQ disease pathologies.

More importantly, we reveal in this study that PQBP5 is deprived to polyQ protein inclusion bodies (consistently to sequestration story). PQBP5 is mislocalized from functional space (nucleoli) to polyQ foci. It means that the situation under polyQ disease is similar to the situation under KD experiment. Like KO experiments, depletion of PQBP5 leads to loss and marked deformity of nucleolus in polyQ diseases, as expected.

So in our idea, the former part and the latter part of this paper should be and are connected, and other reviewers who had raised the same issue of “connection” accepted our previous explanation in R1 revision.

We summarized abovementioned motivation of our group (our research question) in a brief paragraph, and placed the new paragraph at the top of Discussion of R2 manuscript. Also in a sentence of Introduction where we improved our description so that readers can understand more easily our original hypothesis to screen PQBPs.

In revision R1, we wrote that dehydration occurs in old people and those affected by diabetes, in response to the requests from reviewers. Dehydration is a factor to modify neurodegeneration, but it will not be the main factor for the explanation of “connection” we think, as reviewer #3 pointed out.

2.:

my previous comment:

The motivation to study polyQ proteins is a bit more obvious as

PQBP5 is a polyQ binding protein. Yet, how does PQBP5 contribute to polyQ diseases? Are there mutations in PQBP5 that affect disease onset and pathology? This is not clear and it is hard to interpret the data presented here how they relate to polyQ diseases.

answer by the authors:

>>> Again, I would repeat the same explanation. The “sequestration hypothesis” is one of the most popular and still surviving theories in the field of polyQ diseases, and it could be also the case in some other neurodegenerative diseases. Sequestration caused deprivation of PQBP5 from nucleolus, which is equivalent to KO or KD of PQBP5 in regards of the nucleolar function. Therefore, it is easy to speculate, if PQBP5 interacts with polyQ disease proteins, the sequestration of PQBP5 to immobile or aggregated foci (or inclusion body) will reduce the amount of PQBP5 in its functional region in cell (nucleolus), and it will lose the fundamental function of PQBP5 to maintain and anchor nucleolus. We explained this line of mechanistic insight in the first section of the second part (polyQ part) in Result.

As far as we know there is one paper suggesting relationship between PQBP5 and disease, and we referred the paper in Introduction. The relationship between human degenerative diseases and PQBP5 has not been reported so far. It would be an open question.

Such thing will occur, we believe. When we published impairment of DNA damage repair in polyQ pathology based on basic science experiments (Qi et al, Nat Cell Biol 2007; Enokido et al, JCB 2010) there had been no clinical data, while after 10

years it is now common knowledge that SNPs or mutations of DNA damage repair genes affect disease onset and pathology, as the reviewer may know well.

answer from me:

The authors again failed to see my point. I specifically asked for known links between PQBP5 and polyQ such as mutations, GWAS studies etc. Why do they not address this question in the manuscript and quote here in the rebuttal letter where exactly they addressed my question. Because to me this has not been addressed and a response like "while after 10 years it is now common knowledge that SNPs or mutations of DNA damage repair genes affect disease onset and pathology, as the reviewer may know well." is not acceptable.

>>> In line 119 – 123 of our previous R1 revision manuscript, we described about GWAS study, and wrote that there is some information about the link of PQBP5 SNP to prostate cancer but there is no sufficient information about neurodegeneration. We had checked multiple papers of GWAS study on polyQ diseases but those papers did not mention on PQBP5.

However, we apologize the reviewer for our mistake that this sentence (line 119 - 123) was not changed to blue color (instead changed to gray color, in some PCs it looks black letters). We are also sorry that in rebuttal letter of R1 revision, we did not mention about our response related to GWAS study.

In R2, in addition to such previous description in the text, we performed more intensive in-house analysis of previously published GWAS data linked to PQBP5 and checked whether the PQBP5-

SNPs have been reported in relevance to polyQ diseases. Here, we checked raw data of published papers relevant to GWAS studies of polyQ diseases. Despite of these efforts, we could not find reported association of SNPs of PQBP5/NOL10 gene with polyQ diseases. Moreover, there is also no obvious record regarding SCA1 or polyQ diseases in the page of NOL10 of GWAS Catalog (<https://www.ebi.ac.uk/gwas/genes/NOL10>).

However, there are 46,231 human SNPs tagged by PQBP5/NOL10 gene in NCBI SNP database (<https://www.ncbi.nlm.nih.gov/snp/?term=NOL10>), and we could not confirm whether all those SNPs were tested for all polyQ diseases in published 140 papers, and we recognized that only a small number of GWAS studies such as ref 35-38 have been reported in relevance to polyQ diseases.

Therefore in conclusion, though we could not find supportive evidences in published GWAS data, we also cannot say those GWAS studies completely excluded all PQBP5/NOL10-SNPs (46,231) from all polyQ diseases or other neurodegenerative diseases, and the current lack of association at the SNP (or gene) level does not exclude the possibility that PQBP5 is involved in polyQ pathology. Moreover, involvement of PQBP5 in polyQ diseases via other mechanisms at RNA, protein, protein interaction, protein modification, and/or protein degradation levels is possible, even if there was no association in SNPs.

We added description about these new searches in Introduction of R2 manuscript.

3.:

my previous comment:

They end the section of the polyQ proteins (lines 337-339) with a statement: "Interestingly, normal and mutant Atxn1 and Htt proteins were found to bind similarly to PQBP5. Therefore, subsequent processes, such as degradation of the complexes with mutant proteins would contribute to the reduction of PQBP5"

First, how do the authors interpret the data that PQBP5 interacts with non-pathogenic and pathogenic polyQ proteins alike?

Second, do PQBP5 protein levels change upon polyQ expression?

answer by the authors:

>>> We did not stress on the main explanation, "sequestration" here. Even if PQBP5 binds to normal polyQ proteins, (Full sentence in previous rebuttal letter) Even if PQBP5 binds to normal polyQ proteins, it is not sequestered because no insoluble or less mobile inclusion body exists with normal polyQ proteins. After the first explanation, we describe the second explanation by protein degradation after interaction of mutant polyQ proteins and PQBP5.

my response:

what do you mean by "even if PQBP5 binds..." you stated that PQBP5 binds to non-pathogenic and pathogenic polyQ proteins

>>> Yes, PQBP5 binds to non-pathogenic and pathogenic polyQ proteins. In R1 revision, we wrote the reasons in the full sentence.

In the case of mutant polyQ proteins, PQBP5 is trapped to insoluble and less- or non-mobile mutant polyQ proteins, and PQBP5 is mislocalized from nucleus to polyQ foci. In the case of normal polyQ proteins, PQBP5 can move and return from normal polyQ proteins, which are soluble and movable.

So, though PQBP5 binds to normal and mutant proteins, the resultant effects on the mobility of PQBP5 are different. Also PQBP5-mutant polyQ complex is degraded, and PQBP5 is decreased.

Reviewer #4 (Remarks to the Author):

The authors responded to my criticisms of the initial submission well. In particular, I appreciate the statistical analysis of the high speed AFM. As a result, I would support publication based on my expertise.

>>> Thank you very much for kind evaluation and great efforts for our paper. We deeply thank reviewer #4.

Reviewer #5 (Remarks to the Author):

In this manuscript Jin et al describe the nucleolar localisation of PQBP5, identify the C terminus of the protein as being intrinsically disordered and suggest that the protein is key to maintaining the structure of the nucleolus. They also suggests it acts as an anchor for re-assembly of nucleolar structures.

>>> Thank you very much for kind evaluation.

The authors addressed some of the major concerns of reviewer 2 by adding more text to the introduction, results and discussion and citing more papers. However, their additional text required more focus and detail. Especially in relation to previous studies on nucleolar morphology under stress and the role of nucleolar stress in neurodegeneration.

>>>

We used one paragraph (3rd paragraph) in Introduction and three paragraphs in Discussion (2nd, 3rd, 4th paragraphs) to focus on the issue raised by reviewer #5, i.e. the relationship between previous studies on nucleolar morphology under stress and the role of nucleolar stress in neurodegeneration.

Yes, there are some paragraphs in Discussion, which are unrelated to nucleolar morphology under stress or the role of nucleolar stress in neurodegeneration. However, these paragraphs were added in response to the comments of other reviewers in R1 revision, and we believe they are necessary. We kept the paragraphs related to RNA because these are responses to previous comments from other reviewers in R1 revision. We kept the paragraph regarding proteasome-dependent degradation and LLPS for reviewer #1. Also the last paragraph is our response to reviewer #4 and we need to keep it.

Meanwhile, we integrated two redundant paragraphs into one paragraph and shortened the descriptions to show the focus in such paragraphs. We also improved the clarity of these paragraphs by English editing.

They did clarify the specific points reviewer 2 had about the data and included all quantification details as suggested.

>>> Thank you very much for kind evaluation.

“...complete depletion of PQBP5 resulted in the absence of all nucleoli”. This phrase was removed but the statement that PQBP5 is the master regulator of nucleoli remained in the discussion.

>>> We could not find the phrase “PQBP5 is the master regulator of nucleoli” in Discussion of R1 manuscript. However, reviewer #5 might be concerned about some sentences in Discussion.

Therefore, we changed as follows.

1) “Without PQBP5, the other nucleolar proteins could not assemble to form the nucleolus, and PQBP5 was found to function as an anchor for the reassembly of other nucleolar proteins after osmotic stress.”

was changed to

“Without PQBP5, other nucleolar proteins such as fibrillarin and nucelolin could not assemble to form the nucleolus, and PQBP5 was found to function as an anchor for the reassembly of fibrillarin and nucelolin after osmotic stress.

2) “Presumably, this protein could constitute the skeleton of the nucleolus due to its relatively high biophysical stability observed in

thermal shift assay and so on, and functions as an anchor for the reassembly of nucleolar proteins after nucleolar stress.”

was changed to

“Presumably, this protein could constitute the skeleton of the nucleolus due to its relatively high biophysical stability under thermal and osmotic changes, and could function as an anchor for the reassembly of some nucleolar proteins after nucleolar stress.”

“Given that PQBP5 was originally discovered as a binding protein to the polyQ tract sequence²⁵, the relative dominance of PQBP5 among IDPs in nucleolus maintenance was suggestive for how PQBP5 is implicated in polyQ disease pathology.”

was deleted.

Inhibition of Pol I-driven transcription will cause relocalisation of nucleolar proteins into the nucleoplasm/cytoplasm and loss of nucleolar integrity. Therefore, it may be that the consequences of depleting PQBP5 are due to loss of 47S transcription, which could be independent from a structural role. Given that that PQBP5 binds RNA, the effects of PQBP5 loss on 47S transcription should be confirmed.

>>> Thank you very much for the critical comment. The reviewer suggested that 47S rRNA might mediate the effect of PQBP5 depletion. In other words, it is the question whether PQBP5 affects nucleolus directly by itself or indirectly via 47S.

Following suggestion of reviewer #5, we performed new experiments (new Sup Figure 12) and addressed this issue. We checked the expression of 47S rRNA, PQBP5 protein level and GAPDH protein level simultaneously. After 48 hours of shRNA transfection when nucleolus was changed, PQBP5 was decreased but 47S rRNA remained at normal level. After 72 hours, both PQBP5 and 47S rRNA were decreased. The results indicated that PQBP5 affects nucleolus directly but not indirectly via 47S rRNA.

The DAPI channel was not shown in the images in figure 2 and so, it was difficult to see what happened to nuclei and nucleoli (using loss of DAPI staining as a marker for nucleoli rather than Fibrillarin and nucleolin which are altered in stress).

>>> Thank you very much for the critical comment. We rechecked DAPI stains in knockdown experiments (Figure 2) and in osmotic stress experiments (Figure 3), and added panels of DAPI stains to Figure 2 and 3. Thanks to the reviewer #5, we obtained interesting findings.

First, as shown in new Sup Figure 11, non-DAPI regions (DAPI-negative area) in nucleus are not always nucleolus. Non-DAPI regions surrounded by high DAPI signal rings correspond more frequently to nucleoli, but such expectation was still not completely true. Therefore, we concluded expectation/definition of nucleoli by DAPI staining is inaccurate.

With this knowledge, we observed changes of DAPI stains reflecting higher chromatin structure in PQBP5 knockdown and osmotic stresses. Interestingly, though nucleolus disappeared, patterns of DAPI signals were unchanged. This result is very interesting, and

suggesting chromatin structure-dependent nucleolus formation, which is consistent with the current view on nucleolus (for instance reviewed by Schöfer and Weipoltshammer, *Histochem Cell Biol* 2018).

Schöfer C, Weipoltshammer K. Nucleolus and chromatin. *Histochem Cell Biol*. 2018 Sep;150(3):209-225. doi: 10.1007/s00418-018-1696-3. Epub 2018 Jul 25. PMID: 30046888; PMCID: PMC6096769.

** See Nature Portfolio's author and referees' website at www.nature.com/authors for information about policies, services and author benefits.

This email has been sent through the Springer Nature Tracking System NY-610A-NPG&MTS

Confidentiality Statement: □ □ *This e-mail is confidential and subject to copyright. Any unauthorised use or disclosure of its contents is prohibited. If you have received this email in error please notify our Manuscript Tracking System Helpdesk team at <http://platformsupport.nature.com> .*

Details of the confidentiality and pre-publicity policy may be found here <http://www.nature.com/authors/policies/confidentiality.html>

Privacy Policy | Update Profile □ **DISCLAIMER:** This e-mail is confidential and should not be used by anyone who is not the original intended recipient. If you have received this e-mail in error please inform the sender and delete it from your mailbox or any other storage mechanism. Springer Nature America, Inc. does not accept liability for any statements made which are clearly the

sender's own and not expressly made on behalf of Springer Nature America, Inc. or one of their agents.

Please note that neither Springer Nature America, Inc. or any of its agents accept any responsibility for viruses that may be contained in this e-mail or its attachments and it is your responsibility to scan the e-mail and attachments (if any).

REVIEWERS' COMMENTS

Reviewer #3 (Remarks to the Author):

I appreciate the more detailed response by the authors and would support now the publication of the manuscript.

Reviewer #5 (Remarks to the Author):

The revised manuscript has addressed all my concerns and I support its publication.

I thank the authors for addressing my previous comments on 47S transcription and DAPI staining. Their new finding that 47S transcription does not change at 48h post PQBP5 depletion when nucleolar structure is altered, but drops dramatically at 72h post depletion, adds a lot of support to their conclusion that PQBP5 plays a role in structuring the nucleolus. Therefore, it may be useful to add a sentence covering the effects on 47S to the main text.

REVIEWERS' COMMENTS

Reviewer #3 (Remarks to the Author):

I appreciate the more detailed response by the authors and would support now the publication of the manuscript.

>>> Thank you very much for kind evaluation.

Reviewer #5 (Remarks to the Author):

The revised manuscript has addressed all my concerns and I support its publication.

I thank the authors for addressing my previous comments on 47S transcription and DAPI staining. Their new finding that 47S transcription does not change at 48h post PQBP5 depletion when nucleolar structure is altered, but drops dramatically at 72h post depletion, adds a lot of support to their conclusion that PQBP5 plays a role in structuring the nucleolus. Therefore, it may be useful to add a sentence covering the effects on 47S to the main text.

>>> Thank you very much for kind evaluation.

>>> Following the advice, we added one sentence covering the result to the first section of Discussion.